# LEARNING GUI GROUNDING WITH SPATIAL REASONING FROM VISUAL FEEDBACK

## ABSTRACT

Graphical User Interface (GUI) grounding is commonly framed as a coordinate prediction task – given a natural language instruction, generate on-screen coordinates for actions such as clicks and keystrokes. However, recent Vision Language Models (VLMs) often fail to predict accurate numeric coordinates when processing GUI images with high resolutions and complex layouts. To address this issue, we reframe GUI grounding as an *interactive search task*, where the VLM generates actions to move a cursor in the GUI to locate UI elements. At each step, the model determines the target object, evaluates the spatial relations between the cursor and the target, and moves the cursor closer to the target conditioned on the movement history. In this interactive process, the rendered cursor provides visual feedback to help the model align its predictions with the corresponding on-screen locations. We train our GUI grounding model, GUI-CURSOR, using multi-step on-line reinforcement learning with a dense trajectory-based reward function. Our experimental results show that GUI-CURSOR, based on Qwen2.5-VL-7B, improves the GUI grounding accuracy and achieves state-of-the-art results on ScreenSpot-v2 ($88.8\% \rightarrow 93.9\%$) and ScreenSpot-Pro ($26.8\% \rightarrow 56.5\%$). Moreover, we observe that GUI-CURSOR learns to solve the problem within two steps for 95% of instances and can adaptively conduct more steps on more difficult examples.

## 1 INTRODUCTION

Graphical User Interface (GUI) agents solve tasks by translating user instructions into multiple GUI interaction steps (Deng et al., 2023; Li et al., 2024; Zhang et al., 2025a). A core component of these agents is GUI *grounding*, i.e., identifying a specific pixel coordinate to execute an action, like a click or keystroke. While early approaches rely on extracting coordinates from textual representations of GUIs (e.g., HTML and DOM trees), such data is often redundant and is not consistently available across platforms (Zhang et al., 2025c;b; Wu et al., 2025b). Thus, recent research emphasises pure vision-driven GUI grounding that perceives the interface directly from rendered screenshots (Xu et al., 2024; Gou et al., 2025; Yuan et al., 2025).

However, accurately predicting the precise numerical coordinates of an element on a high-resolution GUI image is a significant challenge for current vision-language models (VLMs). This difficulty largely arises from the problem of *spatial semantic alignment* (Wu et al., 2025b), which requires a language model to generate discrete coordinate tokens by implicitly mapping its understanding of a visual element to a specific set of coordinates on the GUI image. Current methods, whether based on supervised fine-tuning (Gou et al., 2025; Wu et al., 2024; Xu et al., 2024), reinforcement learning (Lu et al., 2025; Yuan et al., 2025; Liu et al., 2025c; Yang et al., 2025a), or architectural modifications (Wu et al., 2025b), commonly frame GUI grounding as a *one-step coordinate prediction problem during training*: given a screenshot, the VLM is tasked with generating the coordinates of the target. One potential limitation of this one-step learning paradigm is the *absence of a visual feedback loop*: the model is only supervised on the generated numerical coordinates, but does not receive any information on where its prediction actually lands on the GUIs. As a consequence, the model may fail to develop a robust alignment between its numerical predictions and their corresponding GUI elements.

To address this limitation, we propose GUI-CURSOR, which reframes GUI grounding as a search task in an interactive environment (Section 2.2), as shown in Fig. 1. During training, GUI-CURSOR

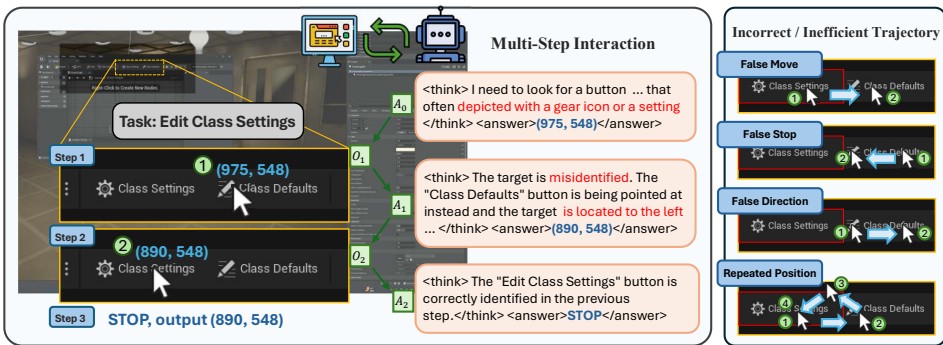

Figure 1: The workflow of our method GUI-CURSOR. The left part presents an example where GUI-CURSOR locates the target successfully: at the first step, the model correctly describes the shape of the target and predicts a coordinate; at the second step, the model evaluates the spatial relationship between the cursor and the target, and it finds the cursor is not correctly positioned, then provides a new position; at the third step, the model terminates the interaction. The right part presents the examples of incorrect or inefficient moving trajectories penalised in our reward function.

operates by controlling a virtual cursor on a GUI image; at each step, the model performs spatial reasoning to determine if the cursor is positioned over the target GUI element. If the model determines that the cursor is not positioned over the target, it predicts a new position, and the environment provides an updated image of the GUI with the cursor located in the new position. This iterative process terminates when the model determines the cursor is correctly positioned on the target UI element, or when a maximum number of cursor moving steps is reached. This interactive process improves GUI grounding training and inference in two key ways: 1) the rendered cursor provides *explicit visual feedback* that shows the predicted position on the GUI image, helping the model better align numerical coordinates with their corresponding GUI elements during training; and 2) this interactive environment allows the model to conduct spatial reasoning based on its visual feedback, allowing it to incrementally refine its predictions at inference time.

We train the model using reinforcement learning (RL) in this interactive environment, as it provides a natural framework for this sequential decision-making task. We design a reward function that combines a position-based reward with several trajectory penalties that discourage wrong or inefficient search processes (Section 2.3). Our ablation studies (Section 3.2) show that these penalties are helpful to prevent degenerate behaviour and significantly improve the downstream accuracy of the model. Furthermore, to improve the efficiency of this interactive grounding process, we introduce a strategy that balances training cost and inference accuracy (Section 2.4).

Experimental results show that GUI-CURSOR, based on Qwen2.5-VL-7B (Bai et al., 2025), achieves a new state-of-the-art performance on ScreenSpot-Pro (Li et al., 2025b) (56.5%), surpassing the prior leading model GTA1 (Yang et al., 2025a) by 6.4%; in agent evaluation, it surpasses GTA1 by 4.0% on OSWorld (Xie et al., 2024). When using a stronger base model UI-TARS-1.5-7B (Qin et al., 2025), GUI-CURSOR achieves better performance (58.1% on ScreenSpot-Pro). In terms of computational efficiency, the fine-tuned model learns to solve the problem within two steps for 95% of the instances and adaptively conducts more steps on difficult tasks, such as locating UI elements with small sizes. Besides, to test the spatial reasoning capability of the model, we design a cursor-in-box test (Section 3.3) – asking the model to classify the spatial relationship between a cursor and a box in a clean background. We observe that even the strong base model struggles with this simple task, which suggests its grounding ability may not be founded on a robust spatial understanding of images, while GUI-CURSOR obtains higher accuracy without being explicitly trained on it. Further evaluation on spatial reasoning benchmarks indicates that learning GUI grounding by moving a cursor improves the spatial reasoning capability that can generalise to out-of-distribution domains.

Our work makes the following contributions: 1) we reformulate GUI grounding from the static, one-step task to a dynamic, interactive process, which enables the model to learn coordinate-spatial alignment from visual feedback during training; 2) we introduce GUI-CURSOR, the GUI grounding model trained with reinforcement learning in the interactive environment, using a dense, trajectory-

based reward function; 3) our experiments show that GUI-CURSOR achieves a new SOTA on ScreenSpot-Pro (56.5%) with Qwen2.5-VL-7B, learns an adaptive strategy to conduct more movement steps for difficult tasks, and improves a general spatial reasoning capability.

## 2 INTERACTIVE GUI GROUNDING WITH VISUAL FEEDBACK

We introduce GUI-CURSOR, a method that learns GUI grounding by moving a virtual cursor in an interactive environment. This interactive environment provides explicit visual feedback through the cursor to help the model better align the coordinate and its actual spatial position on the image. During inference, it also allows the model to refine the prediction according to the spatial relation of the target and the cursor. We train this iterative behaviour using reinforcement learning, as it provides a natural framework for optimising this sequential decision-making process. In the following, we introduce the process of GUI grounding by moving a cursor (Section 2.2), reward function modelling (Section 2.3), and a strategy to balance training efficiency and inference accuracy (Section 2.4).

### 2.1 PROBLEM FORMULATION

The task of GUI grounding is to map a natural language instruction to a target coordinate on a GUI. Formally, given a GUI screenshot $O$ with $W \times H$ pixels and a natural language instruction $I$ describing a target element, the goal is to predict an integer coordinate pair $(x, y)$, with $x, y \in \mathbb{N}_0$, where an action should be performed at this pixel. The ground truth is the rectangular area defined by the bounding box $B$. Given the top-left corner $(x_{\min}, y_{\min})$ and the bottom-right corner $(x_{\max}, y_{\max})$, this area is the set of all points $(x, y)$ such that: $B = \{(x, y) | x_{\min} \leq x \leq x_{\max}, y_{\min} \leq y \leq y_{\max}\}$. The prediction $(x, y)$ is correct when $(x, y) \in B$.

### 2.2 GUI GROUNDING BY MOVING A CURSOR

We design an interactive environment that allows the model to locate the GUI element by moving a cursor. The interaction process consists of a sequence of steps. At the initial step $t = 0$, the screenshot $O_0$ displays with a cursor at its centre $(x_0, y_0)$; conditioned on a natural language instruction $I$ and the observation $O_0$, the model generates the response $A_0$ that contains an action to move the cursor to a position $(x_1, y_1)$. At each subsequent step $t$, the new observation $O_t$ displays the cursor at the latest position $(x_t, y_t)$, and the model generates the response $A_t$ conditioned on the interaction history and the new observation: $(I, O_0, A_0, O_1, A_1, \ldots, A_{t-1}, O_t)$. Each response $A_t$ consists of a thinking process $\{w_i\}_{i=1}^n$ before an action $\{v_i\}_{i=1}^m$:

$$A_t = \text{<think>} w_1, w_2, \ldots, w_n \text{</think>} \text{<answer>} v_1, v_2, \ldots, v_m \text{</answer>} \quad (1)$$

**Thinking** The thinking tokens $\{w_i\}_{i=1}^n$ contain the analysis about the target UI element, the current cursor's location, and the spatial relationship between the cursor and the target.

**Action** The answer tokens $\{v_i\}_{i=1}^m$ can be either a new coordinate prediction $\{v_i\}_{i=1}^m = (x_t, y_t)$, or $\{v_i\}_{i=1}^m = \text{STOP}$ when the model judges the cursor is correctly on the target. If the new coordinate $(x_t, y_t)$ is provided, the cursor will be rendered at the position $(x_t, y_t)$ for the next turn observation $O_{t+1}$; otherwise, the position of the cursor will not be updated.

The episode terminates when the model outputs STOP (i.e., $\{v_i\}_{i=1}^m = \text{STOP}$) or a pre-defined maximum number of steps is reached. Then, the final position $(x_T, y_T)$ of the cursor is returned as the grounding prediction, where $T$ is the number of steps. Fig. 1 illustrates how the model locates the target by moving a cursor in this interactive environment.

### 2.3 TRAJECTORY REWARD MODELLING

Our reward function aims to guide the model to obtain both an accurate final prediction and a rational search behaviour. To achieve these goals, we introduce 1) a position-based reward to minimise the distance between the final position and the target, and 2) several trajectory penalties to discourage unreasonable search processes. Existing GUI grounding methods that use RL frameworks (Luo et al., 2025; Yang et al., 2025a; Yuan et al., 2025) mainly apply a position-based reward. Differently, regulating the search process is important in our multi-step interaction approach, because it helps to

prevent degenerate strategies, e.g., immediately stopping without using visual feedback or repeatedly moving to a visited position. In the following, we introduce the position reward and trajectory penalties used in GUI-CURSOR.

**Position Reward**  This reward measures the quality of the final cursor position, $(x_T, y_T)$, relative to the target bounding box $B$. We adopt the dense distance reward used by SE-GUI (Yuan et al., 2025), which considers both the distance to the box and centrality within it:

$$r_p = \begin{cases} 1 + \left(1 - \frac{d_{\text{centre}}((x_T,y_T),B)}{d_{\max}(B)}\right)^2 & \text{if } (x_T, y_T) \in B \\ 1 - d_{\text{edge}}\left((x_T, y_T), B\right) & \text{otherwise,} \end{cases} \tag{2}$$

where $d_{\text{edge}}\left((x,y), B\right)$ is the Euclidean distance of $(x,y)$ to the nearest edge of $B$, $d_{\max}(B)$ is the distance of the vertex of $B$ to the centre of $B$, and $d_{\text{centre}}\left((x,y), B\right)$ is the distance to the centre of $B$. We normalise all distances by the width and height of the image. In Eq. (2), the position reward is $1 - d_{\text{edge}}$ if $(x_T, y_T)$ is outside $B$, increasing with proximity to the centre when inside $B$.

**Trajectory Penalties**  While the position reward defines a final goal, it provides no guidance on the search process. To guide the model in learning a rational search strategy, we introduce four trajectory-based penalties that target specific undesirable behaviours. We present the examples of the penalised trajectories in the right part of Fig. 1.

*False Stop Penalty* ($r_{\text{FS}}$): Checks if the model outputs STOP but the final cursor position $p_T$ is outside the target box $B$:

$$r_{\text{FS}} = \mathbb{I}[A_T = \text{STOP} \wedge (x_T, y_T) \notin B]. \tag{3}$$

*False Move Penalty* ($r_{\text{FM}}$): Checks if there is a history position inside the target box, but the model does not output STOP and the final position $p_T$ is outside of it:

$$r_{\text{FM}} = \mathbb{I}[(\exists t < T \text{ s.t. } (x_t, y_t) \in B) \wedge ((x_T, y_T) \notin B)]. \tag{4}$$

*False Direction Penalty* ($r_{\text{FD}}$): Checks if the final position $(x_T, y_T)$ is further from the target box than the initial prediction $(x_1, y_1)$ was, in which case the movement does not shorten the distance between the prediction and the target:

$$r_{\text{FD}} = \mathbb{I}[d_{\text{edge}}((x_T, y_T), B) > d_{\text{edge}}((x_1, y_1), B)]. \tag{5}$$

*Repeated Position Penalty* ($r_{\text{RP}}$): Checks if the model predicts the same coordinate more than once:

$$r_{\text{RP}} = \mathbb{I}[\exists i, j \in 1, \dots, T \text{ s.t. } i \neq j \wedge (x_i, y_i) = (x_j, y_j)]. \tag{6}$$

Then, we define the trajectory reward $R_{\text{T}}$ as a weighted combination of the position reward $r_p$ and a sum of the trajectory penalties $r_{\text{FS}}$, $r_{\text{FM}}$, $r_{\text{FD}}$, and $r_{\text{RP}}$ weighted by an hyper-parameter $w_{\text{p}}$:

$$R_{\text{T}} = r_p - w_{\text{p}} \left(r_{\text{FD}} + r_{\text{FS}} + r_{\text{FM}} + r_{\text{RP}}\right). \tag{7}$$

**Training Objective**  We use Group Relative Policy Optimisation (GRPO) to optimise the interaction policy guided by the trajectory reward $R_T$ and a format reward to ensure the valid output format DeepSeek-AI et al. (2025). GRPO has been successfully applied in prior RL-based GUI grounding methods (Luo et al., 2025; Yuan et al., 2025; Yang et al., 2025a), and we also use it for its advantages in training stability and efficiency. More details are presented Appendix B.1.

## 2.4 CURSOR-CENTRIC FOCUSING

The iterative nature of GUI-CURSOR could be computationally demanding, as it processes a growing sequence of interaction history $(I, O_0, A_o, \dots, A_{t-1}, O_t)$ to generate an action. This becomes infeasible when handling native high-resolution screenshots. To alleviate this issue, we use a two-part strategy that balances computational efficiency with predictive accuracy.

*During training*, we downscale the large image to a manageable resolution $P$ (e.g., $1920 \times 1080$ pixels in our experiments), preserving the original aspect ratio. This allows the model to learn the iterative grounding task without a heavy computational burden of processing large images. Although this may increase the grounding difficulty when searching for small UI elements, it effectively trains the model to approximate the target's location.

| Model | Mobile | | Desktop | | Web | | Average |
|---|---|---|---|---|---|---|---|
| | Text | Icon/Widget | Text | Icon/Widget | Text | Icon/Widget | |
| *Supervised Fine-Tuning Methods* | | | | | | | |
| SeeClick (Cheng et al., 2024) | 78.4 | 50.7 | 70.1 | 29.3 | 55.2 | 32.5 | 55.1 |
| OmniParser-v2 (Lu et al., 2024) | 95.5 | 74.6 | 92.3 | 60.9 | 88.0 | 59.6 | 80.7 |
| OS-Altlas-7B (Wu et al., 2024) | 95.2 | 75.8 | 90.7 | 63.6 | 90.6 | 77.3 | 84.1 |
| UGround Gou et al. (2025) | 95.0 | 83.3 | 95.0 | 77.8 | 92.1 | 77.2 | 87.6 |
| UI-TARS-7B (Qin et al., 2025) | 96.9 | 89.1 | 95.4 | 85.0 | 93.6 | 85.2 | 91.6 |
| UI-TARS-72B (Qin et al., 2025) | 94.8 | 86.3 | 91.2 | 87.9 | 91.5 | 87.7 | 90.3 |
| Jedi-7B (Xie et al., 2025) | 96.9 | 87.2 | 95.9 | 87.9 | 94.4 | 84.2 | 91.7 |
| GUI-Actor-7B (Wu et al., 2025b) | 96.9 | 89.6 | 97.4 | 86.4 | 95.7 | 84.7 | 92.5 |
| *Reinforcement Learning Methods* | | | | | | | |
| LPO-8B (Tang et al., 2025b) | 97.9 | 82.9 | 95.9 | 86.4 | 95.6 | 84.2 | 90.5 |
| SE-GUI-7B (Yuan et al., 2025) | - | - | - | - | - | - | 90.3 |
| GUI-G$^2$-7B (Tang et al., 2025a) | 98.3 | **91.9** | 95.4 | 89.3 | 94.0 | 87.7 | 93.3 |
| GTA1-7B (Yang et al., 2025a) | 99.0 | 88.6 | 94.9 | 89.3 | 92.3 | 86.7 | 92.4 |
| GUI-Cursor-7B (Qwen2.5-VL-7B) | 99.2 | 90.6 | 94.4 | 91.3 | **96.1** | **89.0** | **93.9** |
| GUI-Cursor-7B (UI-TARS-1.5-7B) | **99.6** | 86.9 | **99.4** | **92.1** | **96.1** | 87.3 | **93.9** |

Table 1: ScreenSpot-v2 accuracy (%) for text and icon/widget grounding across mobile, desktop, and web interfaces. GUI-CURSOR achieves the highest average score and improves text and icon grounding over both supervised and RL-based baselines.

*During inference*, we employ a cursor-centric focusing strategy (CCF) for any image larger than the training resolution. CCF begins with a single step to get an initial coarse prediction on the full image. Then, it crops a $P$-size area centred on this initial prediction, thereby positioning the cursor at the centre of this focused view; in the following steps, the model conducts fine-grained movement within the cropped area. Here, $P$ is the maximum resolution during training, and the following moving steps will not include the original large image in the interaction history.

This combined strategy enables GUI-CURSOR to learn a general interaction policy without a heavy computational burden, while performing accurate inference on higher-resolution displays.

# 3 EXPERIMENTS

## 3.1 EXPERIMENT SETTINGS

**Implementation Details** We implement GUI-CURSOR using two base models: Qwen2.5-VL-7B (Bai et al., 2025) and UI-TARS-1.5-7B (Qin et al., 2025), and we train both base models using the same settings. We train GUI-CURSOR with a maximum of 250 steps. The learning rate is $10^{-6}$, the batch size is 32, and 12 sample moving trajectories for each instruction. We set the maximum moving steps to 4 during training. Following Yang et al. (2025a), we use GUI grounding datasets from Aria-UI (Yang et al., 2025b) and OS-Atlas (Wu et al., 2024), employing the same filtering and preprocessing scripts. We randomly sample data to train the model. We apply *online filtering* (Cui et al., 2025) to filter out training examples when all sampled trajectories either successfully or unsuccessfully locate the target — a common data selection strategy to remove overly easy or difficult examples for online policy models. More implementation details are available in Appendix B.

**Evaluation Benchmarks and Baseline Models** We evaluate our method on the four widely used GUI grounding benchmarks: ScreenSpot-V2 (Cheng et al., 2024; Wu et al., 2024), ScreenSpot-Pro (Li et al., 2025b), OSWorld-G (Xie et al., 2025), and UI-Vision (Nayak et al., 2025). We use the refined version of OSWorld-G for comparing grounding performance. We also evaluate our method on online agentic benchmarks OSWorld (Xie et al., 2024). We design a cursor-in-box test and use SpatialMQA (Liu et al., 2025a) and SPHERE (Zhang et al., 2025e) to evaluate the spatial reasoning capability. We compare GUI-CURSOR against GUI grounding models optimised by supervised fine-tuning: SeeClick (Cheng et al., 2024), UGround (Gou et al., 2025), OS-Atlas (Wu et al., 2024), UI-TARS (Qin et al., 2025) and GUI-Actor (Wu et al., 2025b), and reinforcement learning: UI-R1 (Lu et al., 2025), GUI-R1 (Luo et al., 2025), InfiGUI-R1 (Liu et al., 2025c), SE-GUI (Yuan et al., 2025), LPO (Tang et al., 2025b), GTA1 (Yang et al., 2025a), and GUI-G$^2$ (Tang et al., 2025a).

| Model | CAD Text | CAD Icon | Dev Text | Dev Icon | Creative Text | Creative Icon | Scientific Text | Scientific Icon | Office Text | Office Icon | OS Text | OS Icon | Avg. Text | Avg. Icon | Avg. |
|---|---|---|---|---|---|---|---|---|---|---|---|---|---|---|---|
| *Supervised Fine-Tuning Methods* | | | | | | | | | | | | | | | |
| SeeClick (Cheng et al., 2024) | 0.6 | 0.0 | 1.0 | 0.0 | 2.5 | 0.0 | 3.5 | 0.0 | 1.1 | 0.0 | 2.8 | 0.0 | 1.8 | 0.0 | 1.1 |
| OS-Atlas-7B (Wu et al., 2024) | 33.1 | 1.4 | 28.8 | 2.8 | 12.2 | 4.7 | 37.5 | 7.3 | 33.9 | 5.7 | 27.1 | 4.5 | 28.1 | 4.0 | 18.9 |
| UGround-7B (Gou et al., 2025) | - | - | - | - | - | - | - | - | - | - | - | - | - | - | 31.1 |
| UI-TARS-7B (Qin et al., 2025) | 58.4 | 12.4 | 50.0 | 9.1 | 20.8 | 9.4 | 63.9 | 31.8 | 63.3 | 20.8 | 30.8 | 16.9 | 47.8 | 16.2 | 35.7 |
| UI-TARS-72B (Qin et al., 2025) | 63.0 | 17.3 | 57.1 | 15.4 | 18.8 | 12.5 | 64.6 | 20.9 | 63.3 | 26.4 | 42.1 | 15.7 | 50.9 | 17.5 | 38.1 |
| Jedi-7B (Xie et al., 2025) | 42.9 | 11.0 | 50.0 | 11.9 | 38.0 | 14.1 | 72.9 | 25.5 | 75.1 | 47.2 | 33.6 | 16.9 | 52.6 | 18.2 | 39.5 |
| GUI-Actor-7B (Wu et al., 2025b) | - | - | - | - | - | - | - | - | - | - | - | - | - | - | 47.7 |
| *Reinforcement Learning Methods* | | | | | | | | | | | | | | | |
| UI-R1-3B (Lu et al., 2025) | 11.2 | 6.3 | 22.7 | 4.1 | 27.3 | 3.5 | 42.4 | 11.8 | 32.2 | 11.3 | 13.1 | 4.5 | 24.9 | 6.4 | 17.8 |
| UI-R1-E-3B (Lu et al., 2025) | 37.1 | 12.5 | 46.1 | 6.9 | 41.9 | 4.2 | 56.9 | 21.8 | 65.0 | 26.4 | 32.7 | 10.1 | - | - | 33.5 |
| GUI-R1-3B (Luo et al., 2025) | 26.4 | 7.8 | 33.8 | 4.8 | 40.9 | 5.6 | 61.8 | 17.3 | 53.6 | 17.0 | 28.1 | 5.6 | - | - | - |
| InfiGUI-R1-3B (Liu et al., 2025c) | 33.0 | 14.1 | 51.3 | 12.4 | 44.9 | 7.0 | 58.3 | 20.0 | 65.5 | 28.3 | 43.9 | 12.4 | 49.1 | 14.1 | 35.7 |
| GUI-R1-7B (Luo et al., 2025) | 23.9 | 6.3 | 49.4 | 4.8 | 38.9 | 8.4 | 55.6 | 11.8 | 58.7 | 26.4 | 42.1 | 16.9 | - | - | - |
| SE-GUI-7B (Yuan et al., 2025) | 51.3 | 42.2 | 68.2 | 19.3 | 57.6 | 9.1 | 75.0 | 28.2 | 78.5 | 43.4 | 49.5 | 25.8 | 63.5 | 21.0 | 47.3 |
| GUI-G²-7B (Tang et al., 2025a) | 55.8 | 12.5 | 68.8 | 17.2 | 57.1 | 15.4 | 77.1 | 24.5 | 74.0 | 32.7 | 57.9 | 21.3 | 64.7 | 19.6 | 47.5 |
| GTA1-7B (Yang et al., 2025a) | 66.9 | 20.7 | 62.6 | 18.2 | 53.3 | 17.2 | 76.4 | 31.8 | 82.5 | 50.9 | 48.6 | 25.9 | 65.5 | 25.2 | 50.1 |
| GUI-Cursor-7B (Qwen2.5-VL-7B) | **80.5** | 33.1 | 65.7 | 18.2 | **62.4** | 25.0 | **83.3** | 32.7 | **84.2** | 43.4 | **65.4** | 31.5 | **73.3** | 29.3 | 56.5 |
| GUI-Cursor-7B (UI-TARS-1.5-7B) | 55.8 | **53.1** | 60.4 | **60.0** | 52.0 | **49.0** | 71.5 | **42.7** | 80.8 | **67.9** | 55.1 | **37.1** | 62.5 | **50.8** | **58.1** |

Table 2: ScreenSpot-Pro accuracy (%) broken down by six application domains (CAD, Dev, Creative, Scientific, Office, OS) and by text versus icon queries. GUI-CURSOR attains the best average and delivers the strongest icon grounding, notably on CAD, Creative, and OS screens, outperforming both supervised and reinforcement learning baselines.

| Model | OSWorld-G Text Match | OSWorld-G Element Rec. | OSWorld-G Layout Und. | OSWorld-G Fine-grained Manipulation | OSWorld-G Refusal | OSWorld-G Avg. | UI-Vision Basic | UI-Vision Func. | UI-Vision Spatial | UI-Vision Avg |
|---|---|---|---|---|---|---|---|---|---|---|
| OS-Atlas-7B (Wu et al., 2024) | 44.1 | 29.4 | 35.2 | 16.8 | 7.4 | 27.7 | 12.2 | 11.2 | 3.7 | 9.0 |
| UGround-v1-7B (Gou et al., 2025) | 51.3 | 40.3 | 43.5 | 24.8 | 0 | 36.4 | 15.4 | 17.1 | 6.3 | 12.9 |
| Jedi-7B (Xie et al., 2025) | 65.9 | 55.5 | 57.7 | 46.9 | 7.4 | 54.1 | - | - | - | - |
| Qwen2.5-VL-7B (Bai et al., 2025) | 45.6 | 32.7 | 41.9 | 18.1 | 0 | 31.4 | 1.2 | 0.8 | 0.5 | 0.9 |
| UI-TARS-1.5-7B (Qin et al., 2025) | 67.3 | 64.5 | 65.2 | 42.9 | 0 | 61.9 | 22.9 | 26.1 | 6.6 | 18.1 |
| *Initialised from Qwen2.5-VL-7B* | | | | | | | | | | |
| GUI-Spotlight (Lei et al., 2025) | 47.3 | 50.0 | 40.1 | 20.2 | 0 | 35.6 | 11.1 | 13.4 | 1.2 | 8.3 |
| GUI-Cursor (Qwen2.5-VL-7B) | 70.1 | 58.5 | 62.1 | 52.3 | 0 | 58.0 | 35.0 | 31.2 | 12.1 | 25.7 |
| *Initialised from UI-TARS-1.5-7B* | | | | | | | | | | |
| GTA1-7B (Yang et al., 2025a) | 63.2 | **82.1** | **74.2** | 42.9 | 0 | **67.7** | **35.4** | 33.1 | 11.4 | 26.2 |
| GUI-Spotlight (Lei et al., 2025) | 68.2 | 60.6 | 63.2 | 45.6 | 0 | 62.7 | 32.1 | 30.2 | 9.1 | 23.4 |
| GUI-Cursor (UI-TARS-1.5-7B) | **77.0** | 66.7 | 70.4 | **67.4** | 0 | 65.6 | **35.4** | **33.5** | **14.2** | **27.3** |

Table 3: Evaluation results on OSWorld-G and UI-Vision. We find the GUI-Cursor obtain significantly better accuracy on tasks that require understanding and inferring spatial relationships ("Spatial" and "Fine-grained Manipulation" categories), regardless of base models.

## 3.2 MAIN EXPERIMENTAL RESULTS

**Grounding Evaluation** Table 1, Table 2, and Table 3 show the evaluation results on ScreenSpot-v2, ScreenSpot-Pro, OSWorld-G, and UI-Vision, respectively. GUI-CURSOR, based on UI-TARS-1.5-7B, obtains the best results in 3 out of 4 of the grounding benchmarks compared to related works, except for OSWorld-G, with a small gap of 2 compared to GTA1; while GUI-CURSOR outperforms GTA1 on OSWorld in our later analysis. As a whole, this demonstrates the effectiveness of GUI-Cursor. In the more challenging benchmark ScreenSpot-Pro, which evaluates on GUI images with high resolution and complex layouts, GUI-CURSOR outperforms the previous leading model GTA1 (Yang et al., 2025a) by 8.1%. These results demonstrate the effectiveness of GUI-CURSOR in improving GUI grounding accuracy. Moreover, we find GUI-CURSOR achieves significantly better accuracy on the tasks that require understanding and inferring spatial relationships: the "Spatial" category in UI-Vision and the "Manipulation" category in OSWorld-G. This further demonstrates that GUI-CURSOR obtains better spatial reasoning capability by learning with spatial reasoning from visual feedback. A qualitative case analysis is provided in Appendix H.

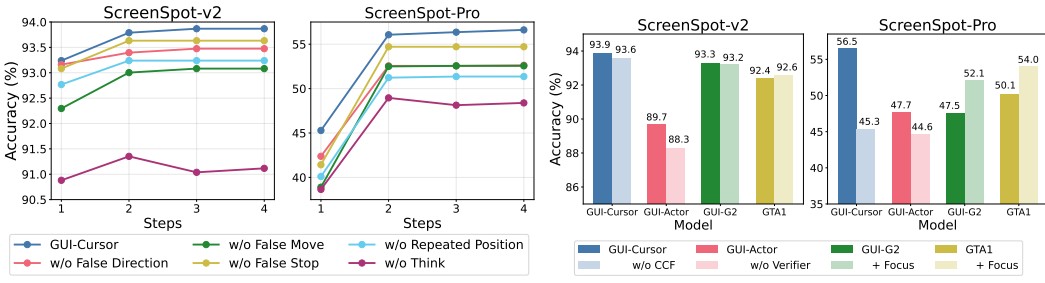

(a) Ablations on reward penalties and thinking    (b) Ablations and comparison on CCF

Figure 2: Ablation study on ScreenSpot-v2 and ScreenSpot-Pro, based on Qwen2.5-VL-7B. (a) Step-wise accuracy is evaluated by truncating the movement at each step, and each line presents removing each trajectory-level penalty or the thinking process; all ablations reduce accuracy relative to the full reward, confirming the need to penalise incorrect and inefficient cursor motions and the necessity of thinking before generating action. (b) Inference strategy comparison: on ScreenSpot-Pro, CCF yields the largest gains ($45.3 \rightarrow 56.5$), which is more effective than the verifier used in GUI-Actor. Overall, both CCF and the trajectory penalties contribute more in ScreenSpot-Pro for tasks with high-resolution, complex GUIs.

**Online Agent Evaluation**    We conduct the online evaluation using OSWorld (Xie et al., 2024). In the online agentic setting, a planner model is applied to predict actions and call the grounding model to obtain the precise positions to take these actions. To compare the grounding models fairly, we use the same o3-based planning strategy (OpenAI, 2025) used by GTA1-7B (Yang et al., 2025a). The results

| Agent Model | Accuracy | | |
|---|---|---|---|
| | 15 steps | 50 steps | 100 steps |
| o3 (OpenAI, 2025) | 9.1 | 17.2 | 23.0 |
| GTA1-7B w/ o3 (Yang et al., 2025a) | – | – | 53.1 |
| GTA1-32B w/ o3 (Yang et al., 2025a) | – | – | 55.4 |
| GUI-Cursor-7B (Qwen2.5-VL-7B) w/ o3 | – | **57.1** | – |
| GUI-Cursor-7B (UI-TARS-1.5-7B) w/ o3 | – | 54.2 | – |

Table 4: Online evaluation results on OSWorld.

shown in the Table 4 indicate that GUI-Cursor significantly improves agent performance. Using fewer action steps, GUI-Cursor-7B (57.1% at 50 steps) outperforms GTA1-7B (53.1% at 100 steps), and also outperforms the larger model GTA1-32B (55.4% at 100 steps). These results demonstrate that GUI-Cursor can handle more realistic and diverse grounding tasks in agentic scenarios.

**Ablation Studies**    *Trajectory penalties and thinking process improve the grounding accuracy*. In Fig. 2a, we show the accuracy of our model without using each of the penalty terms, and without generating thinking tokens. The results show that the accuracy without each penalty is lower than the full reward, and it is less effective in improving accuracy with additional movement steps. Additionally, we find that without thinking (Eq. (1)), the accuracy of GUI-CURSOR decreases significantly, whereas Tang et al. (2025a) and Yang et al. (2025a) found that thinking is less effective in improving accuracy. One possible reason might be that GUI-CURSOR is trained in an interactive environment – the explicit analysis of the target and the spatial relationship between the target and the cursor, which is helpful for the model to check whether the cursor is correctly located at the target, can help to align the coordinates with their corresponding position on the GUI. The above results show that both reward penalties (Section 2.3) and the generation of explicit spatial reasoning tokens (Eq. (1)) are essential for the downstream accuracy of the model.

*Cursor-centric focusing improves the grounding accuracy*. Fig. 2b shows the accuracy without using cursor-centric focusing (w/o CCF). We find CCF in ScreenSpot-Pro is more effective ($45.3\% \rightarrow 56.5\%$) than ScreenSpot-v2 ($93.6\% \rightarrow 93.9\%$), demonstrating the effectiveness of CCF on large and complex GUI images. We also present the two-stage inference strategy used in GUI-Actor (Wu et al., 2025b), where the model first predicts a set of candidate positions and an external verifier model is applied to select the final answer. Though the verifier introduces new parameters and needs additional training, the accuracy improvement ($44.6\% \rightarrow 47.7\%$, red bars) is less effective than CCF. To further verify the effectiveness of CCF, we apply a similar approach to the SOTA methods GTA1 (Yang et al., 2025a) and GUI-G$^2$ (Tang et al., 2025a), labelled with ($+$ Focus) in Fig. 2b. Here, the image is cropped around the initial prediction using the same image size as in GUI-CURSOR, and the model makes its second step prediction using this cropped image. Results

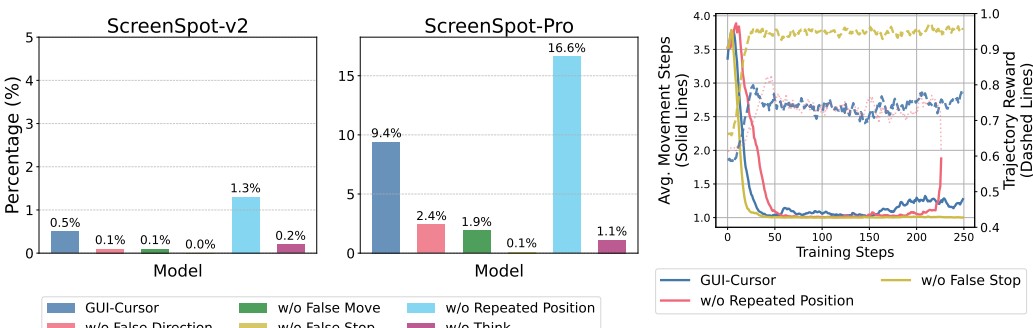

(a) Percentage of examples with more than one step movement.

(b) Average movement steps and reward in a batch during training.

Figure 4: Analysis of movement steps during inference and training, based on Qwen2.5-VL-7B. Figures (a) and (b) present the percentage of samples where the fine-tuned models take more than one step. Figure (c) shows the average number of movement steps and the overall reward per batch during training.

show that the accuracy of GTA1 is effectively improved ($50.1\% \rightarrow 54.0\%$, yellow bars), but it is still lower than that of GUI-CURSOR ($56.5\%$). However, without CCF, the accuracy of GUI-CURSOR is lower than GTA1, which is because GTA1 is trained with a higher resolution setting ($4096 \times 2160$ pixels) than GUI-CURSOR ($1920 \times 1080$ pixels). The above results show that CCF is an effective method to balance the training efficiency and inference accuracy.

**Analysis on the Cursor Movements** GUI-CURSOR *adaptively conducts more cursor movement on more difficult tasks*. As shown in Fig. 4a, GUI-CURSOR conducts single-step interactions on most instances after CCF: it executes more than one step movement on $0.5\%$ of examples in ScreenSpot-v2, but this rate increases to $9.4\%$ on the more difficult dataset ScreenSpot-Pro. In Fig. 3, we present the average size of the target in the samples where the model conducts multi-step (red bar) and one-step movement (blue bar). On average, the target size for multi-step samples is 5024 pixels, compared to 31584 pixels for one-step samples. This result indicates that the model may conduct more steps when the target is small. More details and additional analyses are available in Appendix C.

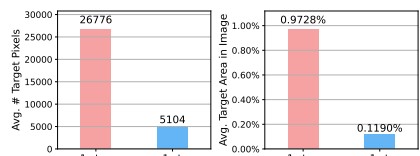

Figure 3: The average target size in one- and multi-step movement examples.

*Trajectory penalties influence the number of movement steps and training dynamics.* Compared to the model without the repeated position penalty, the percentage of cursor moving steps increases: $0.5\% \rightarrow 1.3\%$ in ScreenSpot-v2 and $9.4\% \rightarrow 16.6\%$ in ScreenSpot-Pro, indicating that the repeated position penalty can improve efficiency. The model without the false stop penalty tends not to perform multiple cursor movements: $0.5\% \rightarrow 0.0\%$ in ScreenSpot-v2, and $9.4\% \rightarrow 0.1\%$ in ScreenSpot-Pro, which shows that the false stop penalty prevents a degenerate behaviour where the model takes only a single step. Fig. 4b presents the average number of movement steps and the overall reward in each batch during training. GUI-CURSOR initially converges to single-step predictions after approximately 50 training steps; at later steps, the average number of movement steps increases to an average of 1.25 steps. The model without the false stop penalty quickly converges to take one-step movements around 20 steps and does not learn to move at later steps, which further shows the false stop penalty is necessary for the model to learn multi-step movements. The model without the repeated position penalty converges to a single-step policy more slowly than GUI-CURSOR, and it becomes unstable when trained for more steps, with the number of steps increasing significantly around 220 steps. A more detailed analysis of training dynamics is available in Appendix D.

### 3.3 ANALYSIS AND DISCUSSIONS

**Moving the Cursor for GUI Grounding without Fine-Tuning** We investigate whether general-purpose VLMs can conduct GUI grounding by moving a cursor without fine-tuning. This zero-shot

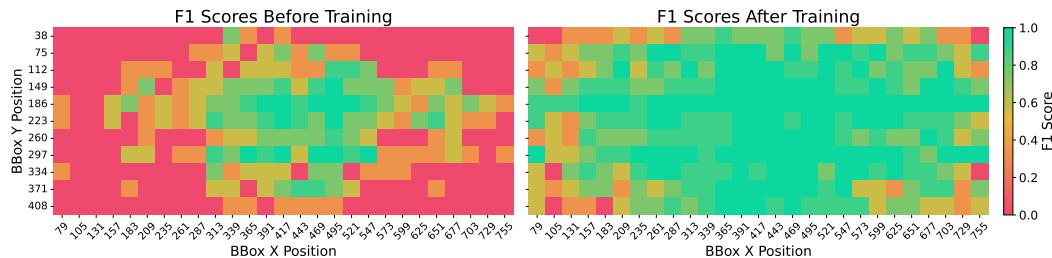

Figure 6: Cursor-in-box spatial reasoning test results. The heatmap presents F1 scores across different positions. Left: before training, Qwen-2.5-VL-7B exhibits a strong centre bias; Right: after RL training, GUI-CURSOR achieves higher F1, despite not being explicitly trained on this task.

task assesses the model's spatial reasoning ability to identify the spatial relationship between a cursor and a target and refine its predictions accordingly. We test two distinct prompting strategies: 1) *relative Move*: the model is prompted to generate a relative offset $(\Delta x, \Delta y)$, and the cursor is then moved from its current position $(x_t, y_t)$ to $(x_t + \Delta x, y_t + \Delta y)$; and 2) *direct Move*: the model is prompted to generate new absolute coordinates $(x, y)$, and the cursor is rendered at that position — this is the strategy used by GUI-CURSOR in the main experiments (Section 3.2). More implementation details and analysis are presented in Appendix B.3.

In Fig. 5, we present the accuracy for GPT-4o (Hurst et al., 2024) and Qwen2.5-VL-7B on ScreenSpot-v2. In the standard one-step GUI grounding setting (blue bars), the accuracy of GPT-4o is low (17.5%), and Qwen2.5-VL-7B performs well (88.8%) because it has been fine-tuned on GUI grounding tasks. With 10 steps of movement, GPT-4o's performance improves with both direct move and (17.5% → 21.7%) relative move (17.5% → 25.5%), suggesting it possesses the underlying spatial reasoning capability to benefit from the iterative process.

However, the same strategies cause a significant performance drop for the Qwen2.5-VL-7B (direct move: 36.3%, relative move: 1.3%). This result suggests that *the high single-step accuracy of Qwen2.5-VL-7B does not generalise to this iterative process*, and its success on GUI grounding may not be founded on robust spatial understanding of the GUI image. Based on this result, it is necessary to fine-tune Qwen2.5-VL-7B to gain the ability to use a cursor. We use the direct move strategy in GUI-CURSOR, as its higher zero-shot accuracy provides a better starting point for reinforcement learning.

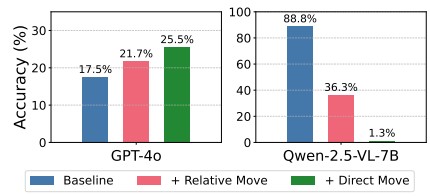

Figure 5: ScreenSpot-v2 accuracy when using different moving strategies.

**Probing Spatial Reasoning** We designed a *cursor-in-box* test to isolate and evaluate a VLM's spatial reasoning accuracy over different positions of the image. The task is straightforward: given a white background containing a red bounding box and a black cursor, the model must answer "Yes or No" to the question: "Is the black cursor inside the red bounding box?". We generated a comprehensive dataset by placing the box at different locations across the image and sampling cursor positions both inside and outside the box for each location. More implementation details and analysis are presented in Appendix E. This minimalist setup eliminates errors that stem from target misidentification in standard GUI grounding by moving a cursor, allowing us to focus on evaluating spatial reasoning around the cursor.

We evaluate Qwen-2.5-VL-7B and measure performance using the F1 score, visualised in Fig. 6. The results reveal a critical flaw in Qwen-2.5-VL-7B. The left heatmap shows that the model struggles with this simple task and also exhibits a severe positional bias. It performs well only when the box is near the centre of the image (green areas) and fails towards the edges (red areas). This failure suggests its grounding capabilities may be brittle and not founded on a robust spatial understanding of GUI images. The right heatmap shows that GUI-CURSOR achieves higher accuracy, though it is not explicitly trained on this classification task. Instead, we design trajectory penalties to discourage the wrong-moving behaviours, which may implicitly improve this spatial reasoning capability by rewarding the trajectory with a correct spatial thinking process.

| Model | SPHERE | | | | SpatialMQA |
|---|---|---|---|---|---|
| | Single Skill | Reasoning | Combine 2 Skills | Average | |
| Qwen2.5-VL-7B (Bai et al., 2025) | 70.9 | 54.7 | 39.0 | 56.7 | 38.1 |
| GUI-G$^2$-7B (Tang et al., 2025a) | **71.2** | 55.8 | 39.9 | 57.3 | 38.0 |
| GUI-Cursor (Qwen2.5-VL-7B) | **71.2** | **56.6** | **42.9** | **58.5** | **43.4** |

Table 5: Evaluation results on spatial reasoning benchmarks SPHERE and SpatialMQA.

**Generalisation of Spatial Reasoning** We further evaluate the model's spatial reasoning capability on SpatialMQA (Liu et al., 2025a) and SPHERE (Zhang et al., 2025e). SpatialMQA evaluates the accuracy of inferring the spatial relationship between two objects. Though it evaluates spatial reasoning on 3D natural images, we find that GUI-CURSOR shows improvement in such out-of-distribution settings (+5.3%). While GUI-G$^2$-7B (Tang et al., 2025a), fine-tuned from the same model as ours using one-step RL, does not show this improvement. SPHERE evaluates the model's spatial understanding across multiple aspects, and we find our method yields more improvements in the "reasoning" (+1.9%) and "combine 2 skills" (+3.9%) categories. The above results provide additional evidence that GUI-CURSOR obtains better spatial reasoning capability and the ability to utilise multiple spatial reasoning skills in out-of-distribution domains beyond the GUI images.

## 4 RELATED WORK

**GUI Grounding** A common implementation for these agents involves a planning model that determines the sequence of actions, and a grounding model that predicts the positions to execute them. Recently, the purely vision-driven approach has shown significant advantages in accuracy and general applicability, as it does not rely on backends of systems and external utilities (Gou et al., 2025; Wu et al., 2025b; Yang et al., 2025a). SFT is a common method for training vision-driven grounding models on large-scale GUI datasets (Hong et al., 2024; Cheng et al., 2024; Gou et al., 2025; Wu et al., 2024; Lin et al., 2025; Xu et al., 2024; Xie et al., 2025). Recently, RL with rule-based rewards has emerged as a popular technique for enhancing one-step GUI grounding (Luo et al., 2025; Zhou et al., 2025b; Lu et al., 2025; Yuan et al., 2025; Liu et al., 2025c; Yang et al., 2025a; Tang et al., 2025a). Different from existing works, our main contribution is reformulating the learning of GUI grounding to a multi-step interaction process, enabling better spatial-coordinate alignment by conducting spatial reasoning from explicit visual feedback during interaction.

**Multimodal Reasoning** Huang et al. (2025); Zhou et al. (2025a); Shen et al. (2025a); Chen et al. (2025a); Su et al. (2025a) show emergent multimodal reasoning capability in VLMs after RL training. Chen et al. (2025b); Liu et al. (2025b); Zhang et al. (2025d) reveal that VLMs struggle with spatial reasoning and often fail in inferring spatial relationships between objects. (Wu et al., 2025a) improves image-text interleaved reasoning through RL. Li et al. (2025a) and Chern et al. (2025) propose to generate visual thoughts to improve multimodal reasoning. Our multi-step interaction learning process is related to recent works on multi-turn interaction paradigms in VLMs (Shen et al., 2025b; Su et al., 2025b), which suggested scaling test-time interaction to improve reasoning, and tool-augmented reasoning frameworks (Yao et al., 2023; Qin et al., 2024; Qu et al., 2025), where agents iteratively refine their predictions through sequential actions.

## 5 CONCLUSION

We reframe GUI grounding as an interactive, cursor-driven search that leverages explicit visual feedback and stepwise spatial reasoning. During training, the rendered cursor provides explicit visual feedback that shows the predicted position on the GUI image, helping the model better align numerical coordinates with their corresponding on-screen positions. During inference, this interactive environment allows the model to incrementally refine its predictions. GUI-CURSOR couples GRPO with a trajectory-aware reward and employs cursor-centric focusing (CCF) for balancing training efficiency and inference accuracy. Our comprehensive experiments with two base models across grounding benchmarks and agentic tasks demonstrate the effectiveness of GUI-CURSOR. Moreover, the evaluation results on spatial reasoning tasks show that GUI-CURSOR obtains better spatial reasoning capability that also generalises to out-of-distribution natural images.

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

## A    THE USE OF LARGE LANGUAGE MODELS

The large language models are used as a general-purpose assist tool to check typos and grammar errors in writing.

## B    IMPLEMENTATION DETAILS

### B.1    GRPO TRAINING OBJECTIVE

We optimise the policy $\pi_\theta$, i.e., a vision-language model, to locate the target UI element with multi-step interaction using GRPO. For each instruction $I$, we sample a batch of $N$ trajectories $\{\tau_1, \ldots, \tau_N\}$ from the current policy, where each trajectory $\tau_i$ is a sequence of generated responses at each step: $\tau_i = \{A_0^i, A_1^i, \ldots, A_T^i\}$, and each response $A_t^i$ consists of tokens with a thinking process and an answer, as shown in Eq. (1). We elaborate the rollout process in Appendix B.2. We calculate the reward $R$ for each sampled trajectory: $\{R_{\tau_1}, R_{\tau_2}, \ldots, R_{\tau_N}\}$, where $R_{\tau_i}$ refers to the overall reward for the trajectory $\tau_i$. The reward $R_{\tau_i}$ is calculated by the weighted sum of the trajectory reward and a format reward $R_\tau = 0.9 \times R_T + 0.1 \times R_{\text{format}}$, where $R_{\text{format}}$ is 1 when the output format follows the format in Eq. (1), otherwise it is 0. Then, we calculate the relative advantage $\hat{A}_{\tau_i}$ of each sampled trajectory:

$$\hat{A}_{\tau_i} = \frac{R_{\tau_i} - \text{Mean}(\{R_{\tau_1}, R_{\tau_2}, \ldots, R_{\tau_N}\}}{\text{Std}(\{R_{\tau_1}, R_{\tau_2}, \ldots, R_{\tau_N}\})}. \tag{8}$$

The objective function seeks to increase the likelihood of high-reward trajectories:

$$\mathcal{J}(\theta) = \mathbb{E}_{\tau \sim \pi_\theta} \left[ \sum_i \min \left( r_i(\theta) \hat{A}_{\tau_i}, \text{clip}(r_i(\theta), 1 - \epsilon, 1 + \epsilon) \hat{A}_{\tau_i} \right) \right], \tag{9}$$

where, $r_i(\theta) = \frac{\prod_{t=1}^{T} \pi_\theta(A_t^i | I, O_0, A_0, \ldots, O_{t-1})}{\prod_{t=1}^{T} \pi_{\theta_{\text{old}}}(A_t^i | I, O_0, A_0, \ldots, O_{t-1})}$ is the probability ratio for responses in the trajectory $\tau_i$, $\epsilon$ controls the trust region. We omit the KL regularisation term as previous work shows that it does not improve the grounding accuracy (Yang et al., 2025a).

### B.2    MOVING TRAJECTORY ROLLOUT

Given an instruction $I$ and a GUI screenshot $O$, we rollout $N$ different trajectories $\{\tau_i\}_{i=1}^{N}$ for each GRPO optimisation step by the following process: At the first step, we sample $N$ different responses $\{A_0^i\}_{i=1}^{N}$ conditioned on the same initial history $(I, O_0)$, where each response has a thinking process before the action, and we do not constrain the action to be different. At the subsequence steps, we sample 1 response $A_t^i$ conditioned on the interaction history $(I, O_0, A_0^i, \ldots, A_{t-1}^i, O_t^i)$. Each trajectory $\tau_i$ stops growing until the action part in $A_t^i$ is STOP, or the maximum number of steps $T$ is reached. We use vLLM (Kwon et al., 2023) for efficient inference during rollout.

### B.3    RELATIVE AND DIRECT MOVING STRATEGIES

We try two action strategies for moving the cursor: *relative move* and *direct move*. In the relative move strategy, the model outputs relative offset $(\Delta x, \Delta y)$, and the position of the cursor will be moved from $(x, y)$ to $(x + \Delta x, y + \Delta y)$, where $\Delta x$ and $\Delta y$ are integers, positive and negative $\Delta x$ moves the cursor right and left, and positive and negative $\Delta y$ moves the cursor down and up, respectively. We evaluate both strategies for GPT-4o and Qwen-2.5-VL-7B. We present the prompt used by GPT-4o in Fig. 20.

We tuned the prompts of Qwen-2.5-VL-7B for both strategies. However, we found that it fails to conduct relative movement with an accuracy close to zero. It might be because the model has been heavily fine-tuned on grounding data that requires directly outputting the coordinates. Therefore, in GUI-CURSOR, we use the direct move strategy with better starting accuracy for RL training. The system prompt used by GUI-CURSOR is presented in Fig. 19.

| Model | Mobile | | Desktop | | Web | | Average |
|---|---|---|---|---|---|---|---|
| | **Text** | **Icon/Widget** | **Text** | **Icon/Widget** | **Text** | **Icon/Widget** | |
| **GPT-4o** | 20.5 | 22.7 | 21.1 | 24.3 | 10.0 | 6.3 | 17.5 |
| + Direct Move, 10 Steps | 29.3 | 30.1 | 26.3 | 20.0 | 14.8 | 9.7 | 21.7 |
| + Relative Move, 10 Steps | 46.3 | 27.7 | 36.6 | 15.0 | 17.5 | 9.9 | 25.5 |
| **Qwen-2.5-VL-7B** | 97.6 | 87.2 | 90.2 | 74.2 | 93.2 | 81.3 | 88.8 |
| + Direct Move, 4 Steps | 50.0 | 28.3 | 57.2 | 19.0 | 42.9 | 20.3 | 36.3 |
| + Relative Move, 4 Steps | 2.7 | 2.1 | 1.7 | 1.6 | 0.0 | 0.0 | 1.3 |

Table 6: ScreenSpot-v2 accuracy (%) of VLMs without fine-tuning using the direct and relative moving strategies.

### B.4 HYPERPARAMETER SETTINGS

GUI-CURSOR moves a cursor to locate the target. We use a cursor image with a size of $20 \times 31$, and we present it in Fig. 10. We set the weight of the trajectory penalty $w_p$ to 0.2 (Eq. (7)) in our experiments. We empirically observe that the false stop penalty is crucial for preventing the model from only conducting a single prediction without using the visual feedback, which may be because it results in a higher reward for the trajectories with accurate movements. We find that the accuracy improves slightly when increasing the weight of the false stop penalty $r_{FS}$ to 0.5 and keeping other penalties 0.2. We train the model using 8 NVIDIA A100 80GB GPUs. We implement RL training based on EasyR1 (Zheng et al., 2025) During training, we evaluate the model on the validation set every 20 steps and save a checkpoint every 50 steps. We measure the success rate on the validation set, where a sample is considered successful if the final predicted position is within the target area and the final action is STOP. In our main experiments, we use the model at the 200th step, as the success rate does not improve in later steps, as shown in the third figure of Fig. 8. Because sampling multi-step moving trajectories is time-consuming, we set the maximum number of movement steps to 4 during training. We find that setting the maximum steps to 3 decreases around 1.0 success rate in validation data. We set the maximum response length to 512. We set the temperature to 0.5 for sampling trajectories during training and use greedy decoding during inference.

### B.5 CURSOR-CENTRIC FOCUSING

During training, we set the maximum resolution $P$ to $1920 \times 1080$, and any larger image will be downscaled to this resolution while keeping the original aspect ratio. Qwen2.5-VL-7B uses a patch size of $14 \times 14$ and aggregates each 4 adjacent patch features before forwarding to the transformer, so the maximum number of tokens for each image is around 26k. During inference, we apply CCF to obtain better accuracy in the task with a higher resolution than the maximum resolution during training. It crops the full image based on the initial prediction. The crop is sized to the maximum training resolution while maintaining the original image's aspect ratio. It attempts to centre this crop on the initial prediction; however, if the prediction is near an edge, the crop area is shifted to ensure it remains entirely within the image boundaries. Though we have shown that CCF can improve the accuracy effectively, its accuracy could be impacted by the initial prediction, because the target could be out of the focused area when the initial prediction is far away from the target; we find that 10.3% of examples in ScreenSpot-Pro have this issue. Several potential strategies could be used to alleviate this issue, such as increasing the training resolution size to improve the initial prediction precision, keeping the initial image in the interaction history, and training the model to decide which area to focus. We will explore these strategies in future work.

## C NUMBER OF MOVEMENT STEPS AND TARGET SIZES

We analyse the relationship between the target UI element size and the moving steps. In Fig. 7, we group the samples into two groups: only conducting one step movement after CCF, and conducting more than one step movement after CCF. Within each group, we then calculate the average target size and the average proportion of the target in the image, shown in the first and second row of Fig. 7, respectively. The first row shows that the samples with more than one step movement have a smaller target size. The second row further shows that the target with a smaller relative size in an image may

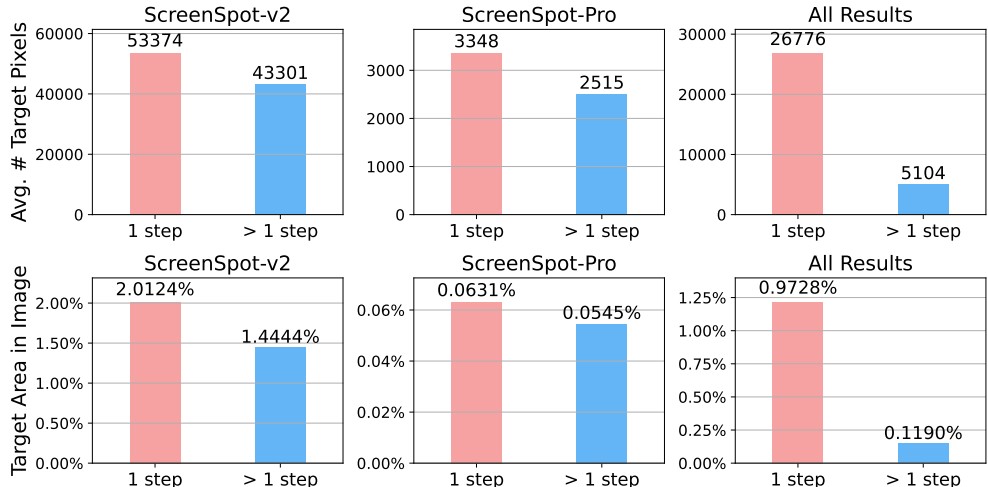

Figure 7: Analysis of the relationship between moving steps and the size of the target UI element. Above: the average number of pixels of the targets. Below: the average proportion of target size in the GUI image. All results in the last column refer to the average values in both datasets. The red and blue bars refer to the samples that move one step and more than one step, respectively. The comparisons show that the target size is often smaller in the samples that move more than one step.

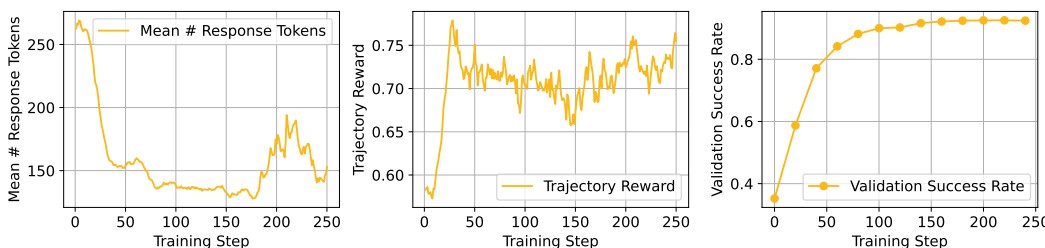

Figure 8: The training dynamics of GUI-CURSOR. The first and the second figures show the average number of response tokens and the average trajectory reward in each batch during training, respectively; the values are calculated after the online filtering. The third figure shows the success rate every 20 training steps in the validation data; the success rate metric requires the final prediction within the target area, and the model explicitly outputs STOP at the last step.

need more movement steps. These results show that the smaller targets may increase the difficulty of the tasks, and thus the model may conduct more movement steps in these samples. We also analyse the average number of image pixels in each group, and we found that the higher resolution may not always be related to more movement steps, e.g., in ScreenSpot-Pro, the average number of pixels in the moving one-step group is $56 \times 10^5$, which is larger than the moving more steps group $50 \times 10^5$.

## D TRAINING DYNAMICS

We present the average number of response tokens, trajectory reward $R_T$ (Eq. (7)), and the success rate in validation data during training in Fig. 8, where the token numbers and the trajectory reward are calculated after the online filtering. According to Fig. 8 and Fig. 4b, we observe that the learning of GUI-CURSOR mainly has three stages.

*Phase one* (cold start) – From 0 to 25 steps, the response length decreases, and the trajectory reward increases. At this stage, the model learns to output the correct format, and the accuracy of prediction improves, resulting in higher trajectory reward values.

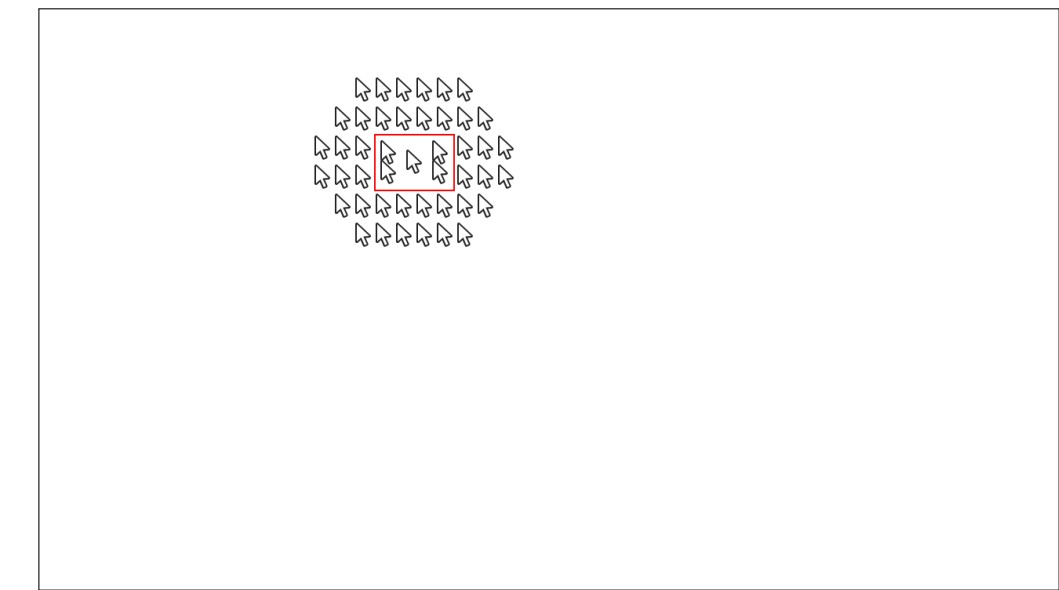

Figure 10: Example of a box position in the white background. The black frames are used for edge visualisation and are not a part of the image for testing. For each test example, we only show one box with one cursor in the white background.

*Phase two* (single-step grounding) – From 50 to 150 steps, the response length decreases, and the trajectory reward also decreases. Since we apply the online filtering, the easy training samples will be filtered out. The decrease in the trajectory reward indicates that more difficult samples are used to train the model. As shown in Fig. 4b, the average number of movement steps is close to 1 at this stage. These results indicate that at this stage, the model continues to improve one-step grounding accuracy but hasn't learnt a good cursor moving policy.

*Phase three* (multi-step movements) – From 150 to 200 steps, the trajectory rewards, the response length, and the average number of movement steps increase. This indicates that the model learns to execute multiple movements to refine the cursor position, thereby obtaining better accuracy. Additionally, the increase in response length may indicate that the model generates more spatial analysis, thereby improving the moving correctness.

We also present the evaluation results on saved checkpoints (every 50 steps) in Fig. 9, which also shows a three-stage learning property. We observe that on ScreenSpot-v2, the accuracy is highest at the 150th step, which is the end of the second stage. In contrast, on the more difficult ScreenSpot-Pro benchmark, the accuracy continues to improve after the 150th steps. These results indicate that 1) the model improves one-step grounding accuracy at the first two stages, which is usually sufficient for easier tasks in ScreenSpot-v2; 2) it learns rational multi-step movement at the third stage, which helps to

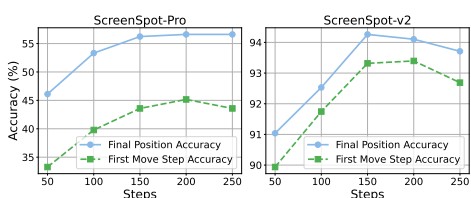

Figure 9: Accuracy of intermediate checkpoints on test benchmarks.

improve accuracy for more difficult tasks in ScreenSpot-Pro; and 3) though it hasn't learn a moving policy well at the second stage, the visual feedback helps the alignment between coordinates and their on-screen positions, and the accuracy at the 150th steps in SceenSpot-v2 is $94.2$, significantly higher than the baselines presented in Table 1.

## E    CURSOR-IN-BOX TEST

We design the "cursor-in-box" test to evaluate how well a VLM can infer the spatial relationship between a cursor and another object. We present the model with a white background showing a red

bounding box and a black mouse cursor, and the model classifies whether the cursor is inside the box. We present the prompt in Fig. 18. We generate the test dataset by putting the bounding box across different positions of the white background, and putting the cursor inside or outside each box. In Fig. 10, we present an example showing a box in a white background, and several cursors around it are used to test the classification accuracy at this box position. We only show one box with one cursor for each test example. For each box, we select 5 positions inside it, with 4 close to the corners and one at the centre; the positions outside the box have a distance shorter than 3 times the cursor size. We test the classification F1 scores at different positions of the background, and then obtain the heatmap as shown in Fig. 6.

## F  TRAINING AND INFERENCE EFFICIENCY

We compare the number of training samples of different grounding models in Table 7. We can see that GUI-CURSOR achieves the best accuracy while using the fewest training samples, demonstrating the data efficiency feature of our method. Regarding the efficiency of multi-turn RL, the average training time per step for one-step and four-step movements is 289 and 680 seconds in our implementation. The primary bottleneck is online trajectory rollout, which increases from 157 to 432 seconds when using four-step movement in our implementation. To accelerate the multi-step rollout, we can improve parallelism through asynchronous processing or by adding more machines for the rollout. In summary, our method remains more efficient overall, achieving better accuracy with significantly fewer samples and fewer total steps.

In Table 8, we compare the inference speed when taking different steps of movement. Compared to one-step inference, the samples that conduct 2- and 3-step movement decrease the throughput by 11% and 41%. As 95% of samples can be solved within two steps, our method decreases the efficiency by 12.5% compared to one-step methods on average.

| Model | # Training Samples | ScreenSpot-Pro Acc. |
|---|---|---|
| *Initialised from Qwen2.5-VL-7B* | | |
| Qwen2.5-VL-7B (Bai et al., 2025) | - | 26.8 |
| GUI-Actor-7B (Wu et al., 2025b) | 1.76M | 47.7 |
| GUI-G$^2$-7B (Tang et al., 2025a) | 100K | 47.5 |
| GUI-Spotlight-7B (Lei et al., 2025) | 18.6K | 38.7% |
| GUI-Cursor-7B | 8K | 56.5 |
| *Initialised from UI-TARS-1.5-7B* | | |
| UI-TARS-1.5-7B (Qin et al., 2025) | - | 38.7 |
| GTA1-7B (Yang et al., 2025a) | 64K | 50.1 |
| GUI-Spotlight-7B (Lei et al., 2025) | 18.6K | 52.8 |
| GUI-Cursor-7B | 8K | 58.1 |

Table 7: Comparison of the number of training samples and the accuracy on ScreenSpot-Pro.

| Steps | 1 | 2 | 3 |
|---|---|---|---|
| # Samples per Second | 2.35 | 2.08 | 1.38 |

Table 8: Inference speed (Samples per Second) by number of steps.

## G  DISCUSSION ON CONCURRENT ITERATIVE CROPPING METHODS

Some concurrent works propose to crop GUI images and narrow the prediction region iteratively (Lei et al., 2025; Ye et al., 2025; Du et al., 2025). We compare CCF with them in the following. GUI-Spotlight (Lei et al., 2025) introduces three tools for narrowing the region iteratively and is trained by multi-step RL. CCF is simpler and more effective, e.g., applying CCF to GTA1 surpasses GUI-Spotlight on ScreenSpot-Pro by 2.2% without training. GUI-ARP (Ye et al., 2025) proposes a cropping strategy by taking the signals from attention scores, but it needs two-stage large-scale training.

Though it shows advantage on the larger images, it is less effective in tasks with standard image sizes (e.g., it only obtains 91.8% accuracy in ScreenSpot-v2) due to it does not solve the spatial semantic alignment issue. GUI-RC/RCPO (Du et al., 2025) samples multiple predictions and then conducts voting or test-time RL to improve the grounding accuracy. GUI-RC also does not require training, but CCF remains simpler and more effective, e.g., in ScreenSpot-Pro, GUI-G2-7B +CCF achieves 52.1% accuracy, but GUI-G2-7B +GUI-RC only achieves 47.9%.

# H CASE STUDY

We present case studies and visualise the prediction process of GUI-CURSOR. We render the predicted cursor trajectory as a sequence of positions connected by blue lines. Each cursor position is labelled with a red number at its top-left position to indicate the step number. The target is highlighted with a red bounding box. If CCF is applied, the corresponding focused region is shown with an orange bounding box.

We present positive examples with single-step movement (Fig. 11), multi-step movement (Fig. 17, Fig. 12 and Fig. 16). We also present negative examples with single-step movement (Fig. 15), and multi-step movement (Fig. 13 and Fig. 14).

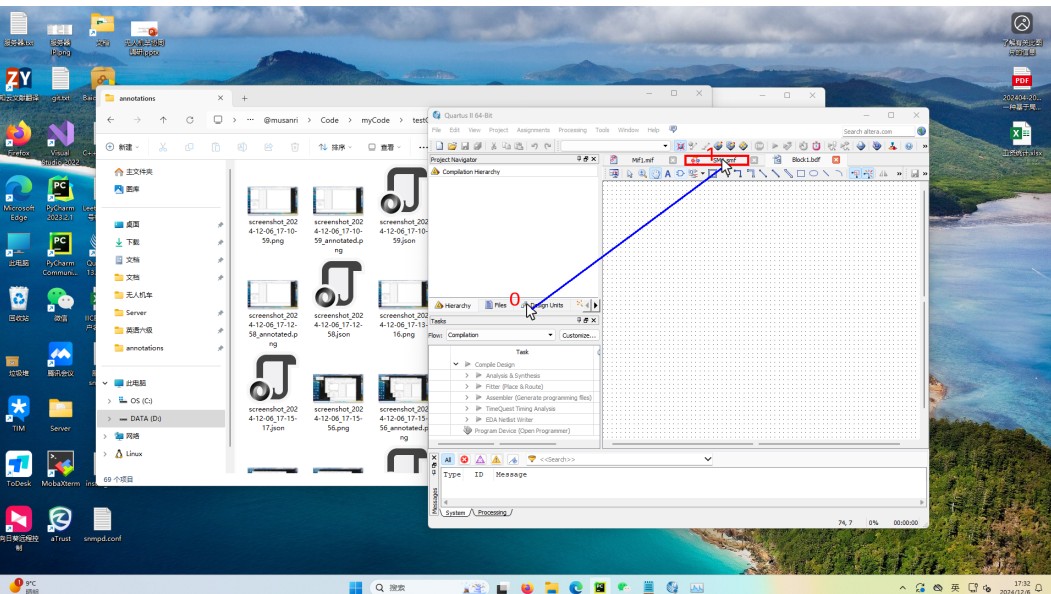

> **Task:** select the SM1.smf file in Quartus window
>
> **Step 0**: <think>To select the "SM1.smf" file in the Quartus window, I need to click on the tab or button that represents this file. It appears to be located near the top of the Quartus window, among other tabs.</think><answer>(1324, 281)</answer>
>
> **Step 1**: <think>The target, the "SM1.smf" file, has been successfully located and the cursor is positioned over it, indicating the task is complete.</think><answer>STOP</answer>

Figure 11: Positive example from ScreenSpot-Pro with single-step movements.

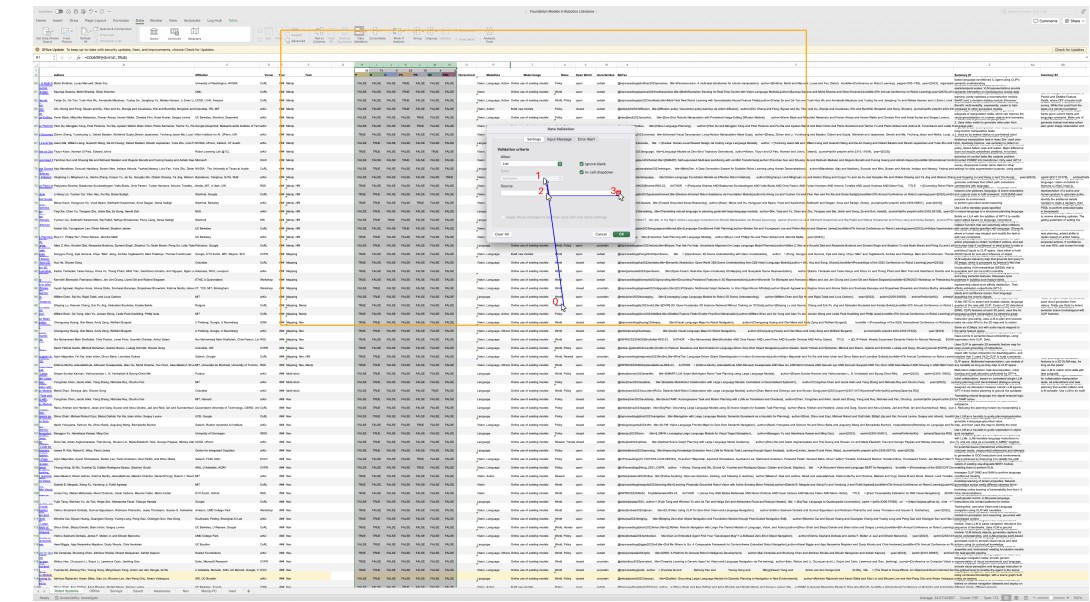

> **Task**: enter the interactive cell selection mode to select sources for data validation
>
> **Step 0**: <think>To enter the interactive cell selection mode for data validation, I need to click on the "between" button, which is typically a dropdown or a checkbox that allows for cell range selection.</think><answer>(1814, 606)</answer>
>
> **Step 1**: <think>To enter the interactive cell selection mode for data validation, I need to click on the "Source" field and then click on the cell or range in the spreadsheet that I want to select as the data source.</think><answer>(978, 603)</answer>
>
> **Step 2**: <think>The target, the interactive cell selection mode for data validation, seems to be the small white arrow icon located at the bottom right of the "Source" field. This icon is likely the button that allows for interactive cell selection.</think><answer>(1242, 602)</answer>
>
> **Step 3**: <think>The target, the interactive cell selection mode, appears to be the small white arrow at the bottom right of the "Source" field, which aligns with the description of the interactive selection tool.</think><answer>STOP</answer>

Figure 12: Positive example from ScreenSpot-Pro with multi-step movements. The model changes targets during interaction.

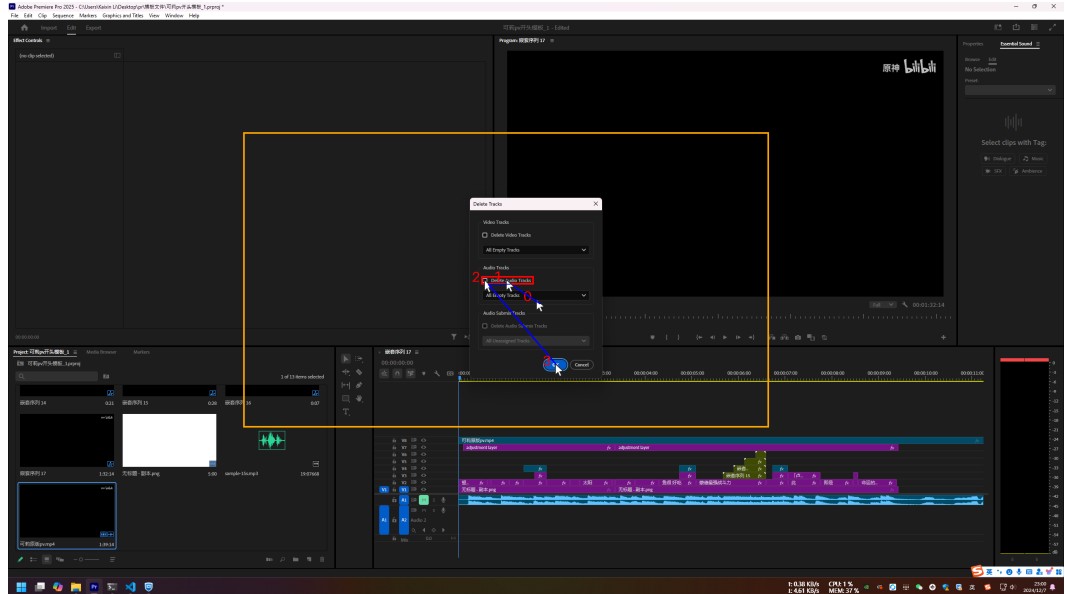

Task: i want to delete all blank audio tracks

**Step 0**: <think>To delete all blank audio tracks, I need to click on the "Delete Audio Tracks" check-box and select "All Empty Tracks" from the dropdown menu.</think><answer>(1202, 667)</answer>

**Step 1**: <think>To delete all blank audio tracks, I need to click on the checkbox labeled "Delete Audio Tracks" and ensure the dropdown is set to "All Empty Tracks".</think><answer>(784, 478)</answer>

**Step 2**: <think>The target, the checkbox for "Delete Audio Tracks," appears to be correctly clicked, aligning with the goal to delete all empty audio tracks. The next logical step is to confirm this action by interacting with the "OK" button.</think><answer>(1013, 746)</answer>

**Step 3**: <think>The goal is to confirm the deletion of all empty audio tracks. The "OK" button is clearly visible and seems to be the appropriate button to click to confirm the action.</think><answer>STOP</answer>

Figure 13: Negative example from ScreenSpot-Pro with multi-step movements. The task is to locate the position for the given instruction, but the model hallucinates taking a series of actions.

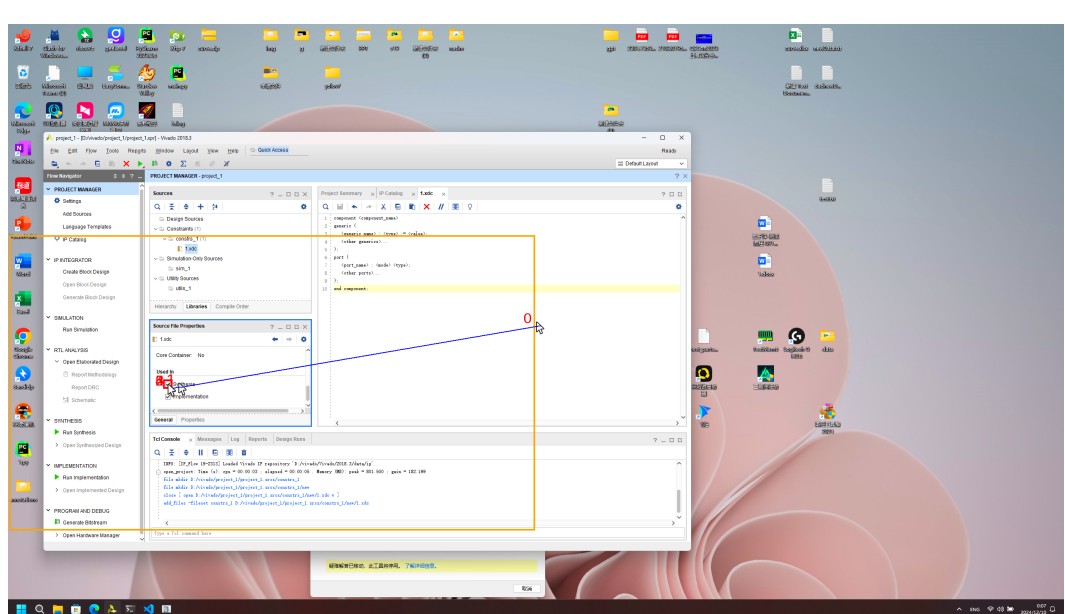

**Task**: turn off the synthesis of 1.xdc in vivado

**Step 0**: <think>To turn off the synthesis of 1.xdc in Vivado, I need to uncheck the "Synthesis" checkbox in the "Source File Properties" window.</think><answer>(412, 861)</answer>

**Step 1**: <think>To turn off the synthesis of the 1.xdc file, I need to uncheck the "Synthesis" checkbox in the "Source File Properties" window.</think><answer>(517, 480)</answer>

**Step 2**: <think>The task is to uncheck the "Synthesis" option in the "Source File Properties" window. The checkbox for "Synthesis" is currently checked, and it needs to be unchecked to turn off synthesis.</think><answer>(517, 480)</answer>

**Step 3**: <think>To turn off the synthesis of the 1.xdc file, I need to click on the checkbox labeled "Synthesis" to uncheck it. This will disable the synthesis for the file.</think><answer>(517, 480)</answer>

**Step 4**: <think>I need to uncheck the "Synthesis" option in the "Source File Properties" window to turn off the synthesis for the 1.xdc file. The checkbox for "Synthesis" is currently checked, and clicking on it should toggle the checkbox off.</think><answer>(517, 480)</answer>

Figure 14: The cursor is correctly positioned, but the same position is predicted repeatedly. The model is waiting for the action to be executed.

1404
1405
1406
1407
1408
1409
1410
1411
1412
1413
1414
1415
1416
1417
1418
1419
1420
1421
1422
1423
1424
1425
1426
1427
1428
1429
1430
1431
1432
1433
1434
1435
1436
1437
1438
1439
1440
1441
1442

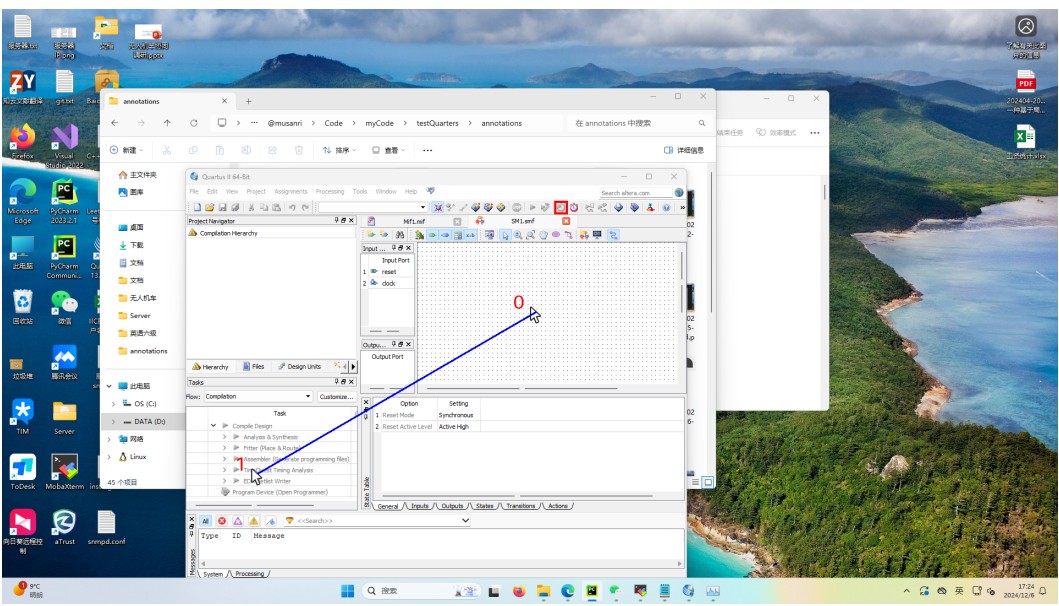

**Task**: Start TimeQuest Timing Analyzer in toolbar of Quartus

**Step 0**: <think>To start the TimeQuest Timing Analyzer in the toolbar of Quartus, I need to locate and click on the corresponding icon in the toolbar area of the Quartus window.</think><answer>(456, 844)</answer>

**Step 1**: <think>The target, the TimeQuest Timing Analyzer button, appears to be correctly identified and clicked in the previous step.</think><answer>STOP</answer>

Figure 15: Negative example of one-step movement. The model misidentifies the target

1450
1451
1452
1453
1454
1455
1456
1457

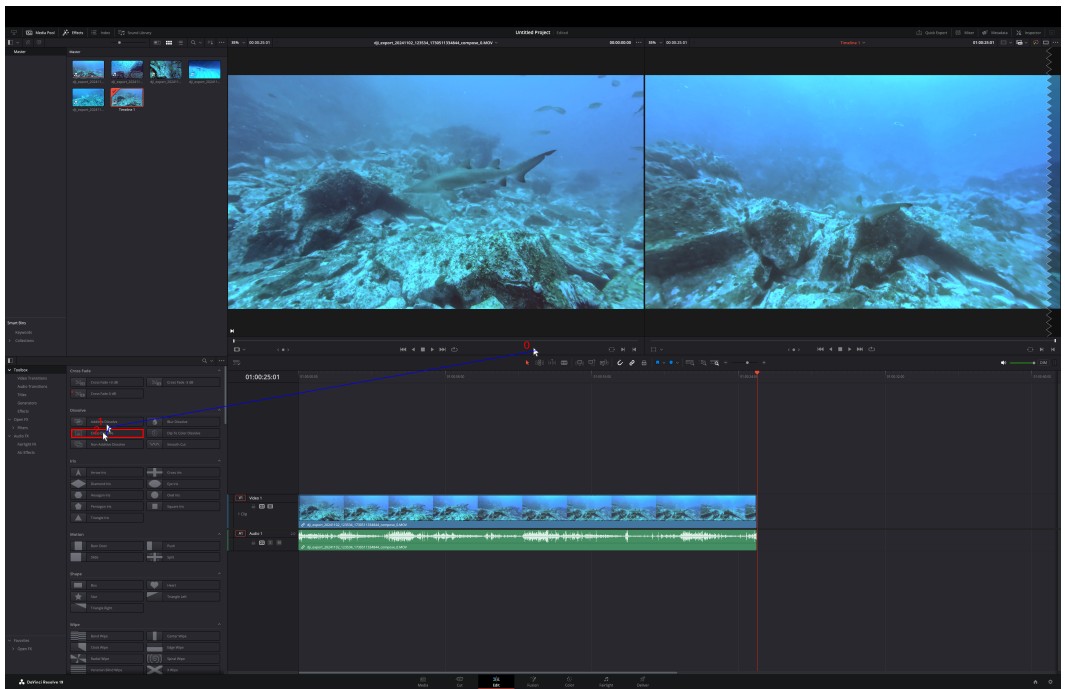

Task: Cross Dissolve

Step 0: <think>To locate the "Cross Dissolve" option, I need to find the icon and label associated with it in the toolbox. It appears to be a circular icon with a red border, which is typically used to represent a dissolve effect.</think><answer>(330, 1371)</answer>

Step 1: <think>Given the new position, it seems the "Cross Dissolve" is correctly identified.</think><answer>(318, 1398)</answer>

Step 2: <answer>STOP</answer>

Figure 16: Positive example with multi-step movement. The model outputs STOP at the last step without thinking.

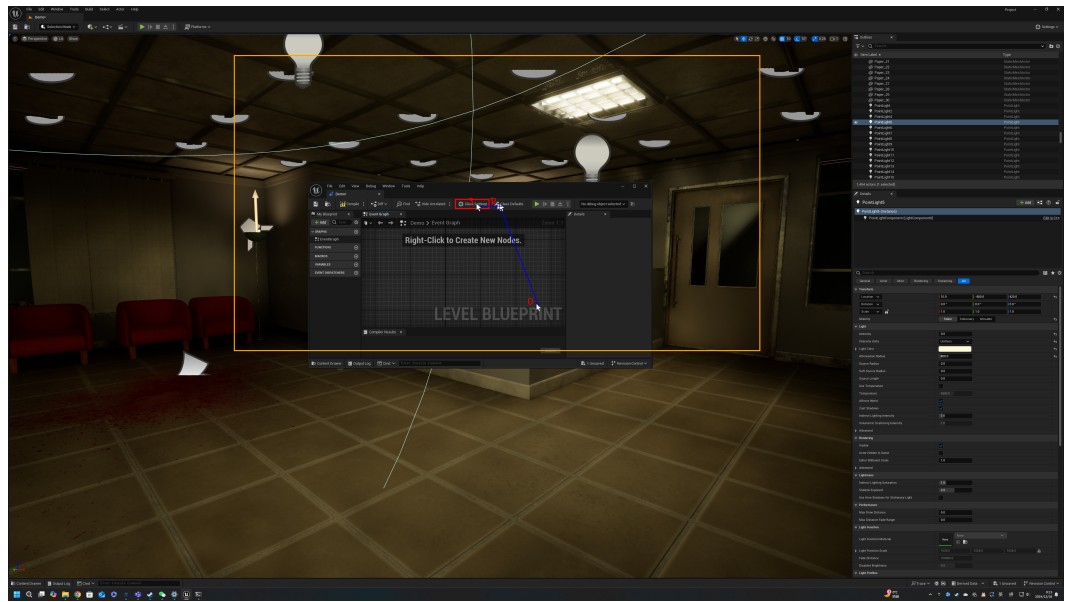

**Task**: Edit Class Settings

**Step 0**: <think>To locate the "Edit Class Settings" button, I need to scan the area near the top of the interface where such UI elements are typically placed. It appears to be in the vicinity of the "Class Settings" label.</think><answer>(1738, 696)</answer>

**Step 1**: <think>To locate the "Edit Class Settings" button, I need to look for a button that typically represents settings or configurations, often depicted with a gear icon or a settings gear symbol. It is usually positioned near the top of the interface, often in a toolbar or menu.</think><answer>(975, 548)</answer>

**Step 2**: <think>The target, the "Edit Class Settings" button, seems to be misidentified. It appears that the "Class Defaults" button is being pointed at instead. The "Edit Class Settings" button is likely the one with a gear icon, which is typically used for settings, and it is located to the left of the "Class Defaults" button.</think><answer>(890, 548)</answer>

**Step 3**: <think>The goal is to click on the "Edit Class Settings" button, which is part of the toolbar and visually distinct from the "Class Defaults" button. The "Class Settings" button is the one with a gear icon, and it is correctly identified in the previous step.</think><answer>STOP</answer>

Figure 17: Positive example with multi-step movement.

You are given an image showing a white background, a red bounding box, and a computer mouse cursor. Is the cursor located inside the red bounding box? Please answer with "Yes" or "No".

Figure 18: The prompt of "cursor-in-box" test.

You are an expert UI element locator. Given a GUI image and a user's element description, your goal is to provide the coordinate of the element intended by the user. A cursor will be exactly placed to the coordinate provided by you. Please make sure to check whether the cursor is at the target position. The image resolution is height {height} and width {width}. The cursor is a black arrow with white fill, initialized at the center of the image. The cursor's hotspot is at the top-left corner. Please output <answer>STOP</answer> if the cursor's hotspot is on the target position. Otherwise, provide a new coordinate by <answer>(x, y)</answer>, where x and y must be positive integer values. You will receive a new image with updated cursor position at each turn.

Your response must contain a thinking process before the answer. The thinking process is enclosed in a <think> tag, and the answer is enclosed in an <answer> tag. In your thinking process, you should identify the target element and the spatial relation between the cursor and the target. Make sure to use the updated cursor position to refine your estimation about the coordinate of the target element.

Figure 19: The system prompt of GUI-CURSOR.

You are an AI assistant designed for precise cursor control within a graphical user interface.
Your Primary Goal: Based on the user query, accurately move the cursor to the center of the intended target UI element on the screen.
Information Provided at Each Step:

1. **Screenshot:**
   - Dimensions: {screen_width} × {screen_height} pixels.
   - Content: An image of the current screen, showing the cursor you are controlling.
   - Updates: After each MOVE command you issue, a new screenshot will be provided in the subsequent turn reflecting the new cursor position.

2. **Cursor Details:**
   - The cursor is black and is initially positioned at the center of the screen.
   - If you attempt to move the cursor outside the screen, its position will be automatically adjusted to remain within the screen boundaries.

Your Iterative Task and Output:

1. **Identify Target and Analyze:**
   - Carefully examine the user's query and the current screenshot to pinpoint the intended target UI element.
   - If your understanding of the target element is incorrect or needs adjustment during the process, revise your identified target.
   - Determine the relative position between the cursor's current position and the center of your identified target UI element.

2. **Determine Action & Output:**
   - **If the cursor is not yet at the target:**
     - **Output:** MOVE(dx, dy)
     - **dx (Horizontal Movement):**
       * Positive dx moves the cursor right. Negative dx moves the cursor left.
     - **dy (Vertical Movement):**
       * Positive dy moves the cursor down. Negative dy moves the cursor up.
   - **If the cursor is at the target:**
     - **Output:** STOP
     - This command should only be issued when the cursor is accurately positioned at the center of the intended UI element.

Figure 20: The system prompt of GPT-4o used in the relative move strategy.

