# OpenReview forum: "Learning GUI Grounding with Spatial Reasoning from Visual Feedback"
_ICLR.cc/2026/Conference — Submitted to ICLR 2026_

### Official Review · Reviewer_JBDJ · 2025-10-28

**Soundness:** 2
**Presentation:** 3
**Contribution:** 2
**Rating:** 4
**Confidence:** 3

**Summary:**

The paper reframes GUI grounding from one-shot coordinate generation to an interactive cursor-moving process. A Qwen2.5‑VL‑7B observes a screenshot with a rendered cursor, reasons about the target and the cursor–target relation, and iteratively issues either a new (x, y) or STOP. Training uses GRPO with a dense distance reward and trajectory penalties. To keep training affordable, the model is trained at 1920×1080 and, at inference, applies cursor‑centric focusing (CCF): one coarse step on the full screen, then crops around the initial click for fine steps. On ScreenSpot‑v2 and ScreenSpot‑Pro, the method reports good performance and shows that most cases finish in 2 steps. The paper also probes spatial reasoning via a simple “cursor‑in‑box” test and argues that interaction plus visual feedback strengthens spatial understanding.

**Strengths:**

- Clear problem reframing. Turning coordinate emission into an interactive search with visual feedback is intuitive and well motivated by spatial–semantic misalignment in patch‑token VLMs. The cursor overlay makes the prediction visible to the model, which is a nice design touch.
- Simple, effective shaping. The four trajectory penalties are targeted at real degeneracies (premature STOP, oscillation, moving the wrong way, repetition). Ablations show each term helps; the gains are especially visible on ScreenSpot‑Pro where large canvases exacerbate these errors.
- A pragmatic efficiency recipe. Training at lower resolution and then zooming near the first guess is a strong “80/20” trick. The authors also apply the same focus idea to baselines, which is thoughtful.

**Weaknesses:**

- The core idea of iterative focus via cropping after an initial guess now appears in multiple concurrent works. GUI‑Spotlight iteratively narrows focus with dedicated crop tools and multi‑turn RL; GUI‑ARP performs adaptive multi‑stage inference with GRPO and attention‑guided crops; GUI‑RC/GUI‑RCPO refine grounding at inference via test‑time consensus/RL. The paper should position CCF and multi‑step cursor moves against these, not just older one‑shot baselines. As it stands, CCF feels close to those “zoom‑in” strategies, and the contribution reduces to (i) rendering a cursor as visual feedback and (ii) a particular set of penalties. [1, 2]
- Outdated SOTA claim. Newer papers report higher ScreenSpot‑Pro accuracy (e.g., GUI‑ARP 60.8% with a 7B backbone), which overtakes the reported 56.5%. The paper should update comparisons and discuss where it still wins (e.g., step efficiency) versus absolute accuracy. [2]
- Missing or under‑cited related work. The submission’s Related Work does not cover GUI‑Spotlight (iterative focus with tools) [1], GUI‑ARP (adaptive region perception and stage control) [2], nor test‑time RL via region consistency (GUI‑RC/GUI‑RCPO) [3]. These are directly relevant to the “interactive narrowing + RL” story and should be acknowledged.
- The cursor‑in‑box probe is neat but synthetic. Given substantial evidence that VLMs struggle with spatial relations, evaluation on public spatial benchmarks (e.g., SpatialMQA, SPHERE) would strengthen the generalization claim [4].

## References
[1] GUI-Spotlight: Adaptive Iterative Focus Refinement for Enhanced GUI Visual Grounding.

[2] GUI-ARP: Enhancing Grounding with Adaptive Region Perception for GUI Agents.

[3] Test-Time Reinforcement Learning for GUI Grounding via Region Consistency.

[4] Can Multimodal Large Language Models Understand Spatial Relations?

**Questions:**

Please refer to the weaknesses.

---

> ### Author Response · Authors · 2025-11-21
>
> Thank you for your review and positive feedback on the clarity of our approach, the simplicity and effectiveness of our reward shaping, and the pragmatic efficiency of CCF. We address your questions and concerns below:
>
> **1. Comparison with Concurrent Work (Weakness 1, Weakness 2, and Weakness 3)**
>
> We appreciate your highlighting the relevance of recent concurrent works, which were developed independently and released shortly before and after the full paper submission deadline (Sep. 24): GUI-Spotlight (Oct. 5, 2025), GUI-ARP (Aug. 7, 2025), and GUI-RC/GUI-RCPO (Sep. 11, 2025). We will update them in the related work section.
>
> GUI-Spotlight, GUI-ARP, and GUI-RC/GUI-RCPO pursue iterative refinement through region cropping or test-time adjustments. GUI-Cursor is complementary to these approaches: it models explicit cursor actions and a STOP decision, which mirrors the control interface of real GUI agents, and is trained with trajectory-level RL penalties tailored for cursor-specific failure modes (false stop, wrong direction, oscillation, repeated positions). Our CCF strategy shares the high-level intuition of progressive focusing, but operates within this cursor-based action space. As shown in Fig. 2(b), similar focusing strategies also improve baselines like GTA1 and GUI-G2, which shows that it is effective and simpler than the concurrent works that require extensive training.
>
> _Outdated SOTA Claim_: We acknowledge the more recent results on the ScreenSpot-Pro leaderboard; the performance of GUI-ARP-7B (60.8%) and Holo1.5-7B (57.9%) does indeed improve over what we reported in our submission in September (56.5%). According to the ICLR 2026 Reviewer Guides [1], these are concurrent works and are not required to be compared with:
>
> > We consider papers contemporaneous if they are published within the last two months. That means, since our full paper deadline is September 24, if a paper was published (i.e., at a peer-reviewed venue) on or after July 24, 2025, authors are not required to compare their own work to that paper. Note that arXiv is not considered a peer-reviewed venue. As such, authors are not required to compare to papers solely on arXiv: they may be excused for not knowing about papers not published in peer-reviewed conference proceedings or journals, which includes papers exclusively available on arXiv.
>
> These works achieve better performance through large-scale training (Holo1.5-7B) or based on a stronger model (GUI-Spotlight is based on UI-TARS-1.5-7B, GUI-ARP-7B is based on GUI-Actor-7B). Our method achieves competitive performance using less training data and a weaker base model. To address your concern, we are currently training GUI-Cursor based on the UI-TARS-1.5-7B model, which provides a stronger starting point. This allows us to obtain better results, and we will include these new results in the updated version of the paper.
>
> [1] ICLR 2026 Reviewer Guide (https://iclr.cc/Conferences/2026/ReviewerGuide)
>
> **2. Spatial Reasoning Evaluation (Weakness 4)**
>
> We agree that moving beyond the synthetic cursor-in-box probe is crucial to validating the model's spatial reasoning abilities. We appreciate your suggestion of evaluating on two public spatial reasoning benchmarks: SpatialMQA and SPHERE. Our experimental results confirm that the interactive cursor-control training in 2D GUI images improves spatial reasoning that can also generalise to natural images. The following is the detailed analysis:
>
> SpatialMQA evaluates the accuracy of inferring the spatial relationship between two objects.
> Though it evaluates spatial reasoning on 3D natural images, we find that GUI-Cursor shows improvement in such out-of-distribution settings. While GUI-G2, fine-tuned from the same model as ours using one-step RL, does not show this improvement. The accuracy is as follows:
>
> | Model | Accuracy |
> | :--- | :--- |
> | Qwen2.5-VL-7B | 38.1% |
> | GUI-Cursor-7B | 43.4% (+ 5.3%) |
> | GUI-G2-7B | 38.0% (- 0.1%) |
> | |
>
>
>
> SPHERE evaluates the model's spatial understanding across multiple aspects, including “single skill” (size, position, distance, counting), “reasoning” (object manipulation and occlusion), and “combine 2 skills” (distance and size, position and counting, etc.). We find our method yields more improvements in the “reasoning” (+1.9%) and “combine 2 skills” (+3.9%) categories. The experimental results are as follows:
>
> | Model | Single Skill | Reasoning | Combine 2 Skills | Average |
> |---|---|---|---|---|
> | Qwen2.5-VL-7B | 70.9% | 54.7% | 39.0% | 56.7% |
> | GUI-Cursor-7B | 71.2% | 56.6% (+1.9%) | 42.9% (+3.9%) | 58.5% (+1.8%) |
> | GUI-G2-7B | 71.2% | 55.8% (+0.9%) | 39.9% (+0.9%) | 57.3% (+0.6%) |
> | |
>
>
> Our method yields significant improvements in the challenging "Reasoning" and "Combine 2 Skills" categories, indicating an enhanced ability to perform complex spatial manipulations and integrate multiple spatial concepts.

---

> ### Author Response · Authors · 2025-11-27
>
> Thank you again for highlighting the concurrent works GUI-Spotlight, GUI-ARP, and GUI-RC/RCPO! We have included a discussion on these concurrent works in the revised submission, Appendix G.
>
> In the following, we clarify how our approach is different from the works you mentioned.
>
> ## Core Idea Clarification
>
> We have to point out that the core idea of GUI-Cursor is not iterative focus via cropping. GUI-Cursor is a complementary approach for the following reason:
>
> 1. **Task Reframing:** GUI-Cursor reframes GUI grounding as a continuous cursor-controlling task, and proposes a set of trajectory penalties tailored to cursor control.
> 2. **Core Objective:** GUI-Cursor’s primary objective is to alleviate the spatial semantic alignment problem, which is achieved by reasoning over visual feedback from cursor usage.
> 3. **Cropping in CCF**: CCF is introduced to balance training efficiency and inference accuracy, and cropping is applied only at inference time when the input size is larger than the training size. In our experiments, we found that this cropping strategy can also be easily applied to other baselines (GTA1 and GUI-G2), significantly improving their accuracy.
>
> The concurrent works primarily focus on narrowing the region by designing better iterative cropping strategies. We included a discussion about these works compared to CCF in the revised paper:
>
> - GUI-Spotlight (released on Oct. 5th, 2025) introduces three tools for narrowing the GUI region iteratively, and is trained via RL. CCF is simpler and more effective; for example, applying CCF to GTA1 yields better results than GUI-Spotlight on ScreenSpot-Pro by 2.2% without any further training.
> - GUI-ARP (released on Sep. 19th, 2025) proposes a cropping strategy by considering the signals from attention scores. Though it shows improvements on the higher resolution GUIs, it is less effective in tasks with standard image sizes (e.g., it only obtains 91.8% in ScreenSpot-v2). A possible explanation is that it does not address the underlying spatial-semantic alignment issue.
> - GUI-RC/RCPO (released on Aug. 8th, 2025) samples multiple predictions and uses voting or test-time RL to improve the grounding accuracy. GUI-RC also does not require training, but CCF remains simpler and more effective; for example, on ScreenSpot-Pro, GUI-G2-7B +CCF achieves higher accuracy (52.1%)  compared to GUI-G2-7B +GUI-RC (47.9%).
>
> GUI-Cursor is a complementary approach, and better cropping methods could be integrated to further improve performance. However, CCF remains the simplest and most effective among these concurrent approaches. We also note that these concurrent works are not required to compare according to the ICLR 2026 Reviewer Guide.

---

> ### Author Response · Authors · 2025-11-28
>
> ## More Comparison on Concurrent Works
>
> We thank the reviewer for mentioning the GUI-ARP-7B (Sep. 19) and Holo1.5-7B (Sep. 15) achieve higher accuracy on ScreenSpot-Pro. As these were released *less than 10 days before the submission deadline* (Sep. 24), they are concurrent work and are not required to be included according to the ICLR 2026 policy. We agree that discussing these concurrent works provides valuable context, and we have included them in the revised submission.
>
> As you suggested, we discuss different aspects of them beyond the accuracy in the following. We agree that a direct comparison of accuracy obscures critical differences in mixed factors rather than the methodology. Below, we clarify these differences and provide new experiments to ensure a fair comparison to GTA1-7B.
>
> **Disentangling data, model, and training pipelines from methodology**
>
> While these concurrent models achieve high accuracy, their performance stems from the mixed factors instead of the methodology itself:
>
> + GUI-APR-7B utilises a stronger base model and a high-quality dataset curated via multi-step filtering and synthesising, trained with a two-phase SFT + RL pipeline.
> + Holo1.5-7B utilises large-scale proprietary high-quality data and a multi-stage SFT + RL pipeline.
>
> These differences in training data, base models, and training strategies prevent a direct, fair comparison of our methodology, i.e., reframing GUI grounding as a continuous cursor-controlling task.
>
> To isolate the effectiveness of our approach, we conducted a new experiment using UI-TARS-1.5-7B—the same base model used by the strong baseline GTA1-7B—and using the same open-source data as GTA1-7B. This allows for a controlled comparison:
>
> | Method | # Training Samples | Training Method | ScreenSpot-v2 Acc. | ScreenSpot-Pro Acc |
> |-|-|-|-|-|
> | Holo1.5 (Qwen2.5-VL-7B) | large scale, proprietary | multi-stage SFT + RL | 93.3 | 57.9 |
> | GUI-ARP (GUI-Actor-7B) | 9.6K + 4.8K | SFT + RL | 91.8 | **60.8** |
> | GTA1-7B ( UI-TARS-1.5-7B | 64K  | Direct RL | 92.4 | 50.1 |
> | GUI-Cursor (Qwen2.5-VL-7B) | 8K | Direct RL | **93.9** | 56.5 |
> | GUI-Cursor (UI-TARS-1.5-7B) | 8K | Direct RL | **93.9** | 58.1 |
> | |
>
> **Conclusion**: Compared to both concurrent works and baselines, GUI-Cursor stands out as the most data-efficient, simple, and effective solution.
>
> + **Data Efficiency**: GUI-Cursor achieves SOTA results on ScreenSpot-v2 and competitive results on ScreenSpot-Pro using only 8K samples, significantly fewer than the concurrent works.
> + **Methodological Superiority**: When using the same base model (UI-TARS-1.5-7B) and the same open source training data, GUI-Cursor (58.1%) outperforms GTA1 (50.1%) by a large margin on ScreenSpot-Pro despite using less data (8K vs 64K).
> + **Simplicity**: GUI-Cursor achieves these results using open datasets and a straightforward direct RL approach without extensive engineering overhead. In contrast, concurrent works rely on multi-stage pipelines and proprietary or heavily curated datasets.

---

### Official Review · Reviewer_opuB · 2025-10-30

**Soundness:** 3
**Presentation:** 3
**Contribution:** 3
**Rating:** 6
**Confidence:** 4

**Summary:**

This paper proposes a solution to the challenge that vision-language models struggle to accurately predict pixel coordinates for GUI elements by reframing GUI grounding as an interactive task. Instead of directly outputting coordinates, the model iteratively moves a virtual cursor on the screen, using visual feedback at each step to refine its position until reaching the target. The model is trained with reinforcement learning using rewards that encourage accurate positioning and penalize inefficient search patterns. GUI-Cursor achieves substantial improvements on ScreenSpot-Pro.

**Strengths:**

S1: Reframing GUI grounding from static coordinate prediction to dynamic cursor-driven search is well-motivated.

S2: The combination of position-based reward with four specific trajectory penalties (false stop, false move, false direction, repeated position) effectively guides learning toward efficient search behaviors.

S3: CCF balances computational constraints during training with accuracy needs during inference.

**Weaknesses:**

W1: The evaluation relies solely on two benchmarks, limiting confidence in generalizability. ScreenSpot-v2 is saturated with baselines already achieving >90%. With substantial gains observed only on ScreenSpot-Pro, it remains unclear whether the multi-step approach generalizes beyond the specific challenges of this benchmark. Evaluation on diverse benchmarks, like OSWorld-G[1] and UI-Vision[2], is necessary to validate that the computational overhead of multi-step interaction provides consistent benefits across different GUI grounding scenarios beyond the ScreenSpot family


W2: The interaction history grows: ($I$, $O_0$,$A_0$, ..., $O_t$). Thus, multi-step interaction accumulates context linearly. With CCF, each $O$ is ~26k tokens.  So, for a hard case that requires 3 steps, the model processes ~78k image tokens and text tokens. Does it slow down inference significantly?

W3: Table 3 reveals that Qwen2.5-VL-7B drops from 88.8% to 36.3% accuracy when using direct cursor movement without fine-tuning. This dramatic failure suggests the base model cannot transfer its existing spatial understanding to cursor control. Consequently, GUI-Cursor's success may primarily reflect learning a new interaction interface (cursor mechanics) rather than improved spatial reasoning. The paper claims to improve "spatial semantic alignment" and "spatial reasoning," but the evidence suggests interface adaptation is the dominant factor. Though the cursor-in-box test shows modest improvements, it uses a highly simplified setting that is far from practical usage.

[1] Xie et al. Scaling Computer-Use Grounding via User Interface Decomposition and Synthesis.

[2] Nayak et al. UI-Vision: A Desktop-centric GUI Benchmark for Visual Perception and Interaction.

**Questions:**

Please see Weaknesses. The reviewer is willing to raise the score if the authors address most, if not all, of the questions above in the Weakness section.

---

> ### Author Response · Authors · 2025-11-21
>
> Thank you for your review and positive feedback on our motivation, reward design and inference strategy! We address your questions and concerns below:
>
> **1. Evaluate on Diverse Benchmarks**
>
> Thank you for suggesting an evaluation on broader benchmarks. We have now evaluated our method on UI-Vision and OSWorld-G. The results confirm that GUI-Cursor yields substantial gains: accuracy improves from 0.7% $\to$ 25.7% on UI-Vision and 31.4% $\to$ 58.0% on OSWorld-G.
>
> We found that the previous leading model GTA1 achieves higher results, which benefits from the stronger base model UI-TARS-1.5. The benefit of initialising from a stronger model has also been verified by the concurrent work GUI-Spotlight [1]. To isolate the efficacy of our method, we are currently training GUI-Cursor on the UI-TARS-1.5. We will update the experiment results soon.
>
> UI-Vision:
>
> | Model | Basic | Functional | Spatial | Average |
> |-|-|-|-|-|
> | Qwen2.5-VL-7B | 1.2% | 0.8% | 0.5% | 0.9% |
> | GUI-Cursor-7B | 35.0% | 31.2% | **12.1%** | 25.7% |
> | GUI-Spotlight (Qwen2.5-VL-7B) | 11.1% | 13.4% | 1.2% | 8.3% |
> | |
> | UI-TARS-1.5-7B | 22.9% | 26.1% | 6.6% | 18.1% |
> | GTA1-7B | 35.4% | 33.1% | 11.4% | 26.2% |
> | GUI-Spotlight (UI-TARS-1.5-7B) | 32.1% | 30.2% | 9.1% | 23.4%
> ||
>
> OSWorld-G:
>
> | Model | Text Match | Element Rec. | Layout Und. | Manipulation | Refusal | Avg. |
> |-|-|-|-|-|-|-|
> | Qwen2.5-VL-7B | 45.6% | 32.7% | 41.9% | 18.1% | 0% | 31.4% |
> | GUI-Cursor-7B | **70.1%** | 58.5% | 62.1% | **52.3%** | 0% | 58.0% |
> | GUI-Spotlight (Qwen2.5-VL-7B) | 47.3% | 50.0% | 40.1% | 20.2% | 0% | 35.6%|
> ||
> | UI-TARS-1.5-7B | 67.3% | 64.5% | 65.2% | 42.9% | 0% | 61.9% |
> | GTA1-7B | 63.2% | 82.1% | 74.2% | 42.9% | 0% | 67.7% |
> | GUI-Spotlight (UI-TARS-1.5-7B) | 68.2% | 60.6% | 63.2% | 45.6% | 0% | 62.7%|
> ||
>
> [1] GUI-Spotlight: Adaptive Iterative Focus Refinement for Enhanced GUI Visual Grounding.
>
> **2. Inference Efficiency**
> > for a hard case that requires 3 steps, the model processes ~78k image tokens and text tokens. Does it slow down inference significantly?
>
> Thanks for asking about the inference speed when processing a potential context! The model needs to process ~78K tokens when the case requires 3 steps. We develop our model using vllm server. Here is the comparison of the throughput when taking different numbers of steps:
>
> | Steps | # Samples per Second |
> | :--- | :--- |
> | 1 | 2.35 |
> | 2 | 2.08 |
> | 3 | 1.38 |
> | |
>
> Compared to one-step inference, the samples that conduct 2- and 3-step movement decrease the throughput by 11% and 41%. As 95% samples can be solved within two steps, our method decreases the efficiency by ~12.5% compared to one-step methods on average. We believe this is an acceptable overhead given the associated performance gains (93.6% $\to$ 93.9% in ScreenSpot-v2, and 45.5% $\to$ 56.5% in ScreenSpot-Pro).
>
> **3. Spatial Reasoning Evaluation**
>
> > The paper claims to improve "spatial semantic alignment" and "spatial reasoning," but the evidence suggests interface adaptation is the dominant factor. the cursor-in-box test uses a highly simplified setting that is far from practical usage.
>
> Thank you for raising the question about spatial reasoning. We agree that the cursor-in-box is synthetic and far from practical usage, and we designed this test because there is no 2D GUI image spatial reasoning benchmark. As suggested by the reviewer JBDJ, we evaluate the spatial reasoning and spatial understanding performance using SpatialMQA and SPHERE.
>
> SpatialMQA evaluates the accuracy of inferring the spatial relationship between two objects.
> Though it evaluates spatial reasoning on 3D and natural images, we find that GUI-Cursor shows improvement in such out-of-distribution settings. While GUI-G2, fine-tuned from the same model as ours using one-step RL, does not show this improvement. The accuracy is as follows:
>
> | Model | Accuracy |
> | :--- | :--- |
> | Qwen2.5-VL-7B | 38.1% |
> | GUI-Cursor-7B | 43.4% (+ 5.3%) |
> | GUI-G2-7B | 38.0% (- 0.1%) |
> | |
>
>
> SPHERE evaluates the model's spatial understanding across multiple aspects, including “single skill” (size, position, distance, counting), “reasoning” (object manipulation and occlusion), and “combine 2 skills” (distance and size, position and counting, etc.). We find our method yields more improvements in the “reasoning” (+1.9%) and “combine 2 skills” (+3.9%) categories. The experimental results are as follows:
>
> | Model | Single Skill | Reasoning | Combine 2 Skills | Average |
> |---|---|---|---|---|
> | Qwen2.5-VL-7B | 70.9% | 54.7% | 39.0% | 56.7% |
> | GUI-Cursor-7B | 71.2% | 56.6% (+1.9%) | 42.9% (+3.9%) | 58.5% (+1.8%) |
> | GUI-G2-7B | 71.2% | 55.8% (+0.9%) | 39.9% (+0.9%) | 57.3% (+0.6%) |
> | |
>
> The above results provide additional evidence that GUI-Cursor obtain better spatial reasoning capability and improves the model's ability to combine multiple spatial reasoning skills in out-of-distribution domains. We will include the above spatial reasoning evaluation results in the updated version of the paper.

---

> > ### Comment · Reviewer_opuB · 2025-11-25
> >
> > Thank the authors for the detailed response.
> >
> > I am convinced by the authors' response to W2 & 3. As for W1, I do want to see the performance gain based on UI-Tars-7b, since the authors introduce a training pipeline, not just a grounding model. I feel it should be effective on the most commonly used model. I am looking forward to the results of that. Also, if the authors can share the results of OS-World mentioned in the response to Reviewer 1NKj, that would be very helpful for me to justify the work's contribution more fairly.

---

> ### Author Response · Authors · 2025-11-26
>
> Thank you for your response! As requested, we obtained evaluation results of GUI-Cursor based on UI-TARS-1.5-7B. The results confirm that our method can be applied to other base models and also show the benefit of initialising from a stronger base model.
>
> Based on UI-TARS-1.5-7B, GUI-Cursor achieves significantly better performance and closes the gap on OSWorld-G. We find that GUI-Cursor achieves substantial improvements in categories that require spatial reasoning: the categories “Spatial” in UI-Vision and “Manipulation” in OSWorld-G, regardless of base models. Here are examples for these categories:
> + “Spatial” example task from UI-Vision: “What element is horizontally to the right of 'display selected console'”?
> + “Manipulation” example task from OSWorld-G: “Select the place between the world ’person’ and the number ’1’”.
>
> _This observation further indicates that GUI-Cursor obtained better spatial reasoning capability in our multi-step RL training method._
>
> ### UI-Vision
>
> | Model | Basic | Functional | Spatial | Average |
> |-|-|-|-|-|
> | Qwen2.5-VL-7B | 1.2% | 0.8% | 0.5% | 0.9% |
> | GUI-Spotlight (Qwen2.5-VL-7B) | 11.1% | 13.4% | 1.2% | 8.3% |
> | GUI-Cursor-7B (Qwen2.5-VL-7B) | 35.0% | 31.2% | **12.1%** | 25.7% |
> | |
> | UI-TARS-1.5-7B | 22.9% | 26.1% | 6.6% | 18.1% |
> | GTA1-7B (UI-TARS-1.5-7B) | 35.4% | 33.1% | 11.4% | 26.2% |
> | GUI-Spotlight (UI-TARS-1.5-7B) | 32.1% | 30.2% | 9.1% | 23.4% |
> | GUI-Cursor-7B (UI-TARS-1.5-7B) | **35.4%** | **33.5%** | **14.2%** | **27.3%** |
> ||
>
> ### OSWorld-G
>
> | Model | Text Match | Element Rec. | Layout Und. | Manipulation | Refusal | Avg. |
> |-|-|-|-|-|-|-|
> | Qwen2.5-VL-7B | 45.6% | 32.7% | 41.9% | 18.1% | 0% | 31.4% |
> | GUI-Spotlight (Qwen2.5-VL-7B) | 47.3% | 50.0% | 40.1% | 20.2% | 0% | 35.6%|
> | GUI-Cursor-7B (Qwen2.5-VL-7B) | **70.1%** | 58.5% | 62.1% | **52.3%** | 0% | 58.0% |
> ||
> | UI-TARS-1.5-7B | 67.3% | 64.5% | 65.2% | 42.9% | 0% | 61.9% |
> | GTA1-7B (UI-TARS-1.5-7B) | 63.2% | **82.1%**| **74.2%** | 42.9% | 0% | **67.7%** |
> | GUI-Spotlight (UI-TARS-1.5-7B) | 68.4% | 60.6% | 63.2% | 45.6% | 0% | 62.7%|
> | GUI-Cursor-7B (UI-TARS-1.5-7B) | **77.0%** | 66.7% | 70.4% | **59.7%** | 0% | 65.6% |
> ||
>
> ### Overall Results on Grounding Benchmarks
>
> We summarise the results of four grounding benchmarks in the table below. GUI-Cursor obtains the best results in 3 out of 4 of the grounding benchmarks compared to related works. Despite a narrow 2% gap compared to GTA1-7B on OS-World-G, GUI-Cursor outperforms GTA1 in the later OSWorld analysis. As a whole, this demonstrates the effectiveness of GUI-Cursor.
>
> | Model | ScreenSpot-v2 | ScreenSpot-Pro | OSWorld-G | UI-Vision |
> |-|-|-|-|-|
> | GTA1-7B (UI-TARS-1.5-7B) | 92.4% | 50.1% | **67.7%** | 26.2% |
> | GUI-Spotlight (Qwen2.5-VL-7B) | - | 38.7% | 35.6% |  8.3% |
> | GUI-Spotlight (UI-TARS-1.5-7B) | - | 52.8% | 62.7% | 23.4% |
> | GUI-Cursor-7B (Qwen2.5-VL-7B) | 93.9% | 56.5% | 58.0% | 25.7% |
> | GUI-Cursor-7B (UI-TARS-1.5-7B) | **93.9%** | **58.1%** | 65.6% | **27.3%** |
> ||
>
>
> ### OSWorld Evaluation
>
> We evaluate GUI-Cursor on the agentic benchmark OSWorld. We use the same o3-based planning strategy used by GTA1-7B for fair comparison. The evaluation results show that using fewer action steps, GUI-Cursor (57.1% at 50 steps) outperforms GTA1-7B (53.1% at 100 steps), and also outperforms the larger model GTA1-32B (55.4% at 100 steps). The results further demonstrate that GUI-Cursor can handle more realistic grounding tasks in more dynamic scenarios.
>
> | Agent Model | Acc. (15 steps) | Acc. (50 steps) | Acc. (100 steps) |
> |-|-|-|-|
> | o3 | 9.1% | 17.2% | 23.0% |
> | UI-TARS-1.5-7B | - | - | 27.4% |
> | GTA1-7B w/ o3 | - | - | 53.1% |
> | GTA1-32B w/ o3 | - | - | 55.4% |
> | GUI-Cursor (Qwen2.5-VL-7B) w/ o3 | - | **57.1%** | - |
> ||

---

### Official Review · Reviewer_1NKj · 2025-10-31

**Soundness:** 3
**Presentation:** 3
**Contribution:** 3
**Rating:** 6
**Confidence:** 4

**Summary:**

This paper proposes GUI-Cursor, a reinforcement learning framework for GUI grounding that reframes one-step coordinate prediction as an multi-step interactive search problem, where the model iteratively moves a virtual cursor to locate UI elements while receiving visual feedback at each step. The key insight is that existing VLMs suffer from spatial-semantic misalignment because they only receive supervision on numerical coordinates without seeing where their predictions actually land on the GUI.
GUI-Cursor addresses this by training with GRPO reinforcement learning using a trajectory-level reward design (including penalties for false direction, false stop, repeated position, etc.) and a Cursor-Centric Focusing (CCF) inference strategy to balance efficiency and accuracy.
Built upon Qwen2.5-VL-7B, GUI-Cursor achieves state-of-the-art results on ScreenSpot-v2 (93.9%) and ScreenSpot-Pro (56.5%, +6.4% over prior best), with the model learning to adaptively use more steps for difficult cases (e.g., small targets).
Additionally, cursor-in-box spatial reasoning test show that GUI-Cursor significantly enhances the model’s ability to understand and reason about spatial relationships between the cursor and visual elements, indicating that reinforcement-driven interaction helps build more robust spatial understanding beyond task-specific grounding.

**Strengths:**

**1. Clear and well-motivated approach:** The reformulation of GUI grounding as an interactive search task with visual feedback is intuitive and addresses a fundamental limitation of existing methods: models receive supervision only on numerical coordinates without seeing where predictions actually land, leading to spatial-semantic misalignment.

**2. Effective inference strategy:** The proposed Cursor-Centric Focusing (CCF) balances computational efficiency and accuracy, particularly on high-resolution, complex GUIs.

**3. Strong empirical results:** GUI-Cursor-7B achieves state-of-the-art performance on both ScreenSpot-v2 (93.9%) and ScreenSpot-Pro (56.5%, +6.4% over prior best).

**4. Comprehensive and insightful analysis:** The paper provides detailed ablations, movement-step analysis, and spatial reasoning diagnostics that clarify how interaction and feedback improve spatial understanding. The cursor-in-box diagnostic test reveals that strong VLMs struggle with basic spatial reasoning and exhibit severe center bias, while interactive training on GUI grounding improves this capability as an emergent property, showing that visual feedback genuinely enhances spatial understanding beyond task-specific performance.

**5. Clear presentation:** The paper is well-written, logically structured, and supported by clear figures and experimental evidence.

**Weaknesses:**

**1. Lack of downstream validation:** The evaluation focuses solely on static grounding benchmarks without demonstrating practical benefits in real GUI agent systems or broader multimodal interaction scenarios.

**2. Title and spatial reasoning scope:**
The paper prominently features “Spatial Reasoning” in its title, but the treatment of this aspect is somewhat limited. The *cursor-in-box* diagnostic test is interesting, but the paper does not provide a deeper analysis of how GUI grounding relates to spatial reasoning or whether this capability can transfer to other spatial reasoning tasks. Additionally, the mechanism by which visual feedback enhances spatial understanding remains underexplored.

**3. Computational cost:** While 95% of examples are solved within 2 steps, the paper does not analyze the computational overhead of processing multiple high-resolution images with growing context windows, or compare inference efficiency against single-step baselines. The trade-off between accuracy gains and computational cost deserves more thorough investigation.

**Questions:**

**1. Ablation on single-step training:** Could the authors provide results for a single-step version of GUI-Cursor trained under identical settings (same base model, GRPO algorithm, training data, and position reward) but constrained to max_steps = 1? This would help isolate the contribution of multi-step interactive learning from other design factors.

**2. Evaluation in interactive agent settings:** While ScreenSpot-v2 and ScreenSpot-Pro are strong static grounding benchmarks, evaluating GUI-Cursor in more realistic interactive agent environments (e.g., *WebArena* and *Multimodal-Mind2Web* for web, *AndroidControl* for mobile, and *OSWorld* for operating system agents) could better demonstrate its applicability to real-world GUI agent tasks.

**3. Comparison with concurrent work:** According to the latest ScreenSpot-Pro leaderboard ([https://gui-agent.github.io/grounding-leaderboard/](https://gui-agent.github.io/grounding-leaderboard)), more recent results report improved performance. For instance, GTA1-7B has been updated to 55.5%, and newer models such as GUI-ARP-7B (60.8%) and Holo1.5-7B (57.9%) achieve higher scores. Could the authors clarify whether these are concurrent submissions and discuss how GUI-Cursor compares to these latest results? If these methods were developed independently around the same time, a brief discussion of their methodological differences would be valuable for the community.

**4. Figure presentation:** In Figure 2(b), some text labels overlap with the bars, which slightly affects readability. Improving the layout or adjusting the legend placement could make the comparison clearer.

---

> ### Author Response · Authors · 2025-11-21
>
> Thank you for your review and positive feedback on the clarity, motivation, strong empirical results, and comprehensive analysis of our approach, including the insightful spatial reasoning diagnostics. We address your questions and concerns below:
>
> **1. Downstream Validation**
> >  Lack of downstream validation: The evaluation focuses solely on static grounding benchmarks without demonstrating practical benefits in real GUI agent systems or broader multimodal interaction scenarios.
>
> Thank you for raising this question! Based on your comment, we started an online evaluation on OSWorld and an offline evaluation on Multimodal-Mind2Web. For OSWorld, we use the same planning method as GTA1 [1], allowing us to directly compare our method with it. For Multimodal-Mind2Web, we use the existing planning results from UGround [2] to fairly compare grounding performance. We are currently running these experiments; we will notify you as soon as they finish and include the new results in the updated paper.
>
> [1] GTA1: GUI Test-time Scaling Agent (https://arxiv.org/abs/2507.05791)
> [2] Navigating the Digital World as Humans Do: Universal Visual Grounding for GUI Agents (https://arxiv.org/abs/2410.05243, ICLR’25)
>
> **2. Spatial Reasoning Evaluation**
> > Title and spatial reasoning scope: The paper prominently features “Spatial Reasoning” in its title, but the treatment of this aspect is somewhat limited. The cursor-in-box diagnostic test is interesting, but the paper does not provide a deeper analysis of how GUI grounding relates to spatial reasoning or whether this capability can transfer to other spatial reasoning tasks. Additionally, the mechanism by which visual feedback enhances spatial understanding remains underexplored.
>
> Thank you for this question – after your question, we evaluated GUI-Cursor on SpatialMQA and SPHERE, two spatial reasoning benchmarks suggested by reviewer JBDJ. We find that GUI-Cursor improves spatial reasoning abilities on general natural images beyond GUI images; it also outperforms the baseline GUI-G2, which is fine-tuned from the same base model Qwen2.5-VL-7B with single-step RL.
>
> SpatialMQA evaluates the accuracy of inferring the spatial relationship between two objects. The accuracy is as follows:
>
> | Model | Accuracy |
> | :--- | :--- |
> | Qwen2.5-VL-7B | 38.1% |
> | GUI-Cursor-7B | 43.4% (+ 5.3%) |
> | GUI-G2-7B | 38.0% (- 0.1%) |
> | |
>
> These results indicate that GUI-Cursor improves the model’s ability to infer spatial relationships between objects in natural images, beyond those in GUI images.
>
> SPHERE evaluates the model's spatial understanding across multiple aspects, including “single skill” (size, position, distance, counting), “reasoning” (object manipulation and occlusion), and “combine 2 skills” (distance and size, position and counting, etc.). We find our method yields significant improvements in the “reasoning” (+1.9%) and “combine 2 skills” (+3.9%) categories. The experimental results are as follows:
>
> | Model | Single Skill | Reasoning | Combine 2 Skills | Average |
> |---|---|---|---|---|
> | Qwen2.5-VL-7B | 70.9% | 54.7% | 39.0% | 56.7% |
> | GUI-Cursor-7B | 71.2% | 56.6% (+1.9%) | 42.9% (+3.9%) | 58.5% (+1.8%) |
> | GUI-G2-7B | 71.2% | 55.8% (+0.9%) | 39.9% (+0.9%) | 57.3% (+0.6%) |
> | |
>
> The above results show that GUI-Cursor yields better spatial reasoning results and improves the model's ability to combine multiple spatial reasoning skills in out-of-distribution domains. It demonstrates that our main idea – reframing GUI grounding as a cursor-controlling task – can benefit both a model’s GUI grounding and spatial reasoning abilities.
>
> We will include the above spatial reasoning evaluation results in the updated version of the paper.
>
> **3. Computational Cost**
> > 3. Computational cost: While 95% of examples are solved within 2 steps, the paper does not analyze the computational overhead of processing multiple high-resolution images with growing context windows, or compare inference efficiency against single-step baselines. The trade-off between accuracy gains and computational cost deserves more thorough investigation.
>
> Thank you for the question about the inference efficiency! We use vllm server for efficient inference and benchmark the throughputs of different steps; the results are shown in the table:
>
> | Steps | # Samples per Second |
> | :--- | :--- |
> | 1 | 2.35 |
> | 2 | 2.08 |
> | 3 | 1.38 |
> | |
>
> Compared to one-step inference, the samples that conduct 2- and 3-step movement decrease the throughput by 11% and 41%. As 95% samples can be solved within two steps, our method decreases the efficiency by ~12.5% compared to one-step methods on average. We believe this is an acceptable overhead given the associated performance gains (93.6% $\to$ 93.9% in ScreenSpot-v2, and 45.5% $\to$ 56.5% in ScreenSpot-Pro).

---

> ### Author Response · Authors · 2025-11-21
>
> **4. Single-Step Ablation**
>
> >  Could the authors provide results for a single-step version of GUI-Cursor trained under identical settings (same base model, GRPO algorithm, training data, and position reward) but constrained to max_steps = 1? This would help isolate the contribution of multi-step interactive learning from other design factors.
>
> Thanks for asking this! We add the ablation study with max_steps=1 during training while maintaining all other settings identical. We find that it results in a significant performance drop, as shown in the table. This aligns with findings from prior work GTA and GUI-G2, which also found that single-step reasoning without visual feedback is less effective.
>
> | Model | ScreenSpot-v2 | ScreenSpot-v2 w/o CCF | ScreenSpot-Pro | ScreenSpot-Pro w/o CCF |
> | :--- | :---: | :---: | :---: | :---: |
> | GUI-Cursor | 93.9% | 93.6%| 56.5% | 45.3% |
> | GUI-Cursor (1-step) | 90.1% | 89.7% | 47.8% | 40.7% |
> | |
>
> **5.  Evaluation in Agent Settings**
> > Experiments are conducted only on GUI grounding benchmarks. It remains unclear whether the proposed method generalizes to broader GUI agent tasks—especially those involving user instructions that do not exactly match GUI elements. Additional experiments on datasets such as AITW or Multimodal-Mind2Web would strengthen the claim that the proposed interactive reasoning approach can handle more realistic and diverse scenarios.
>
> We appreciate the suggestion to evaluate on broader agent tasks. We started the online evaluation on OSWorld and the offline evaluation on Multimodal-Mind2Web. For OSWorld, we use the same planning method as GTA1 [1], allowing us to directly compare our method with it. For Multimodal-Mind2Web, we use the existing planning results from UGround [2] to fairly compare grounding performance. We are currently running these experiments; we will notify you as soon as they finish and include the results in the updated version of the paper.
>
> [1] GTA1: GUI Test-time Scaling Agent (https://arxiv.org/abs/2507.05791)
> [2] Navigating the Digital World as Humans Do: Universal Visual Grounding for GUI Agents (https://arxiv.org/abs/2410.05243, ICLR’25)
>
> **6. Comparison with Concurrent Works**
> >  According to the latest ScreenSpot-Pro leaderboard (https://gui-agent.github.io/grounding-leaderboard/), more recent results report improved performance. For instance, GTA1-7B has been updated to 55.5%, and newer models such as GUI-ARP-7B (60.8%) and Holo1.5-7B (57.9%) achieve higher scores. Could the authors clarify whether these are concurrent submissions and discuss how GUI-Cursor compares to these latest results? If these methods were developed independently around the same time, a brief discussion of their methodological differences would be valuable for the community.
>
>
> We thank the reviewer for highlighting these very recent updates and concurrent works, GUI-ARP [1], GUI-RC/GUI-RCPO [2], Holo1.5 [3]). These papers appeared shortly before our submission. While ICLR 2026 review policy does not require comparisons to works before July 24, 2025 (https://iclr.cc/Conferences/2026/ReviewerGuide), we sincerely appreciate the reviewer drawing our attention to them, and we will incorporate a dedicated subsection in the Related Work to situate GUI-Cursor in this rapidly evolving landscape.
>
> Conceptually, methods such as GUI-Spotlight, GUI-ARP, and GUI-RC/GUI-RCPO pursue iterative refinement through region cropping or test-time adjustments. GUI-Cursor is complementary to these approaches: it models explicit cursor actions and a STOP decision, which mirrors the control interface of real GUI agents, and is trained with trajectory-level RL penalties tailored for cursor-specific failure modes (false stop, wrong direction, oscillation, repeated positions). Our CCF strategy shares the high-level intuition of progressive focusing, but operates within this cursor-based action space. As shown in Fig. 2(b), similar focusing strategies also improve baselines like GTA1 and GUI-G2, which shows that it is effective and simpler than the concurrent works.
>
> [1] (Sep. 19, 2025) GUI-ARP: Enhancing Grounding with Adaptive Region Perception for GUI Agents https://arxiv.org/abs/2509.15532
>
> [2] (Aug. 7, 2025) Test-Time Reinforcement Learning for GUI Grounding via Region Consistency https://arxiv.org/abs/2508.05615
>
> [3] (Sep. 11, 2025) Holo1.5: Foundational Models for Computer Use Agents https://huggingface.co/Hcompany/Holo1.5-7B

---

> ### Author Response · Authors · 2025-11-27
>
> **5. Evaluation in Agent Settings Update**
>
> Thank you again for the suggestion to evaluate in interactive agent settings. We have conducted two additional evaluations: 1) Online evaluation on OSWorld and 2) Offline evaluation on Multimodal-Mind2Web. The detailed results are as follows:
>
> ### OSWorld
>
> We use the same o3-based planning strategy used by GTA1-7B for fair comparison. The results shown in the table below indicate that GUI-Cursor significantly improves agent performance. Using fewer action steps, GUI-Cursor-7B (57.1% at 50 steps) outperforms GTA1-7B (53.1% at 100 steps), and also outperforms the larger model GTA1-32B (55.4% at 100 steps). These results demonstrate that GUI-Cursor can handle more realistic grounding tasks in agentic scenarios.
>
> | Agent Model | Acc. (15 steps)| Acc. (50 steps)| Acc. (100 steps) |
> |-|-|-|-|
> | o3 | 9.1% | 17.2% | 23.0% |
> | UI-TARS-1.5-7B | - | - | 27.4% |
> | GTA1-7B w/ o3 | - | - | 53.1% |
> | GTA1-32B w/ o3 | - | - | 55.4% |
> | GUI-Cursor-7B w/ o3 | - | **57.1%** | - |
> ||
>
> ### Multimodal-Mind2Web
>
> We use the planning results provided by UGround to compare the grounding performance fairly. As shown in the table (below), GUI-Cursor achieves the highest average accuracy, but we observe that performance differences are small across all models for this benchmark. Furthermore, we manually checked the results and found that the majority of errors stem from the planner (GPT-4o) rather than the grounding module. So to improve results, one should use a better planner such as GPT-5.
>
> | Method | Cross-Domain Acc. | Cross-Task Acc. | Cross-Website Acc. | Avg Acc. |
> |-|-|-|-|-|
> | UGround-V1-7B | 48.5% | **50.7%** | 48.1% | 49.1% |
> | GUI-G2 | 48.5% | 49.1% | 48.2% | 48.6% |
> | GTA1-7B | 48.7% | 49.3% | 47.9% | 48.6% |
> | GUI-Cursor-7B | **48.9%** | 50.2% | **49.0%** | **49.4%** |
> ||
>
> In summary, the results on the agentic benchmarks OSWorld and Multimodal-Mind2Web demonstrate that GUI-Cursor can handle grounding tasks in more realistic interactive agent environments.

---

### Official Review · Reviewer_MH8z · 2025-11-01

**Soundness:** 3
**Presentation:** 3
**Contribution:** 2
**Rating:** 4
**Confidence:** 4

**Summary:**

The authors tackle the mislocalization problem in existing visual GUI agents and propose an interactive visual reasoning approach that evaluates spatial relations between the cursor and the target. In this framework, the rendered cursor serves as visual feedback for vision-language models. The proposed method is validated on standard GUI grounding benchmarks.

**Strengths:**

- The draft is well-organized and clearly written.
- The intuition behind the proposed method is solid and straightforward.
- The authors provide well-designed ablation studies that help deepen the understanding of their approach.

**Weaknesses:**

- One concern lies in the novelty of the proposed method. In the broader vision domain — such as object grounding and visual question answering — many prior works have already utilized initial predictions as visual feedback, often through bounding boxes or check marks. This work appears to be a straightforward application of that idea to GUI grounding tasks, and the discussion does not sufficiently clarify how it differs from or advances beyond previous literature in general vision contexts.

- There are also concerns about the trade-off between training cost and performance. The current draft is not convincing regarding the practicality of this method. Please provide a detailed comparison of training efficiency with existing supervised fine-tuning (SFT) approaches such as GUI-Actor-7B or RL approaches GTA1-7B to better understand the improvements. Given the recent trend toward large-scale pretraining in GUI agents, such a Pareto analysis is particularly important.

- Experiments are conducted only on GUI grounding benchmarks. It remains unclear whether the proposed method generalizes to broader GUI agent tasks—especially those involving user instructions that do not exactly match GUI elements. Additional experiments on datasets such as AITW or Multimodal-Mind2Web would strengthen the claim that the proposed interactive reasoning approach can handle more realistic and diverse scenarios.

**Questions:**

- It is unclear whether reinforcement learning is necessary for this method. Could the proposed approach also function effectively without additional training, for example in a test-time scaling or supervised fine-tuning (SFT) setting?

---

> ### Author Response · Authors · 2025-11-21
>
> Thank you for your comments and feedback, and we appreciate that you believe our work is clear and well-organised, intuitively sound, and with well-designed ablation studies. We address your concerns below:
>
> **Question 1**
> > In the broader vision domain — such as object grounding and visual question answering — many prior works have already utilized initial predictions as visual feedback, often through bounding boxes or check marks.
>
> Thank you for mentioning the broader context of vision tasks. Could you share the specific prior works you are referring to? We tried to be as exhaustive as possible in our related work section, and we are happy to improve it further.
>
> > the discussion does not sufficiently clarify how it differs from or advances beyond previous literature in general vision contexts.
>
> Our main idea is to reframe GUI grounding as a continuous cursor-control problem. We find that this reframing allows the model to better align its prediction with its physical position by using spatial reasoning from visual feedback during training; thus, the model’s predictions become more accurate and require fewer movement steps during training. Moreover, the cursor-in-box test (Figure 6) and spatial reasoning evaluations on SpatialMQA and SPHERE (suggested by reviewer JBDJ) show that our model's spatial reasoning capabilities improve in out-of-distribution domain (SpatialMQA: 38.1% $\rightarrow$ 43.4%, SPHERE-Reasoning: 54.7% $\rightarrow$ 56.6%, SPHERE-Combine2Skill: 39% $\rightarrow$ 42.9%),  demonstrating the benefits of our task reframing.
>
> **Question 2**
> > There are also concerns about the trade-off between training cost and performance. The current draft is not convincing regarding the practicality of this method. Please provide a detailed comparison of training efficiency with existing supervised fine-tuning (SFT) approaches such as GUI-Actor-7B or RL approaches, GTA1-7B to better understand the improvements. Given the recent trend toward large-scale pretraining in GUI agents, such a Pareto analysis is particularly important.
>
>
> Thank you for this question! Regarding training costs, our method uses significantly fewer training samples than both existing SFT- and RL-based approaches. In the following, we provide a detailed comparison.
>
> For the model initialised from Qwen-2.5-VL-7B, we compare the accuracy of the best available SFT-based method (GUI-Actor-7B), the best RL-based method (GUI-G2-7B), and the most related concurrent work (GUI-Spotlight-7B, suggested by reviewer JBDJ):
>
>
> | Model | # Training Samples | ScreenSpot-Pro Acc. |
> | :--- | :--- | :--- |
> | Qwen2.5-VL-7B | - | 26.8% |
> | GUI-Actor-7B | 1.76M | 47.7% |
> | GUI-G2-7B | 100K | 47.5% |
> | GUI-Spotligh-7B (Qwen2.5-VL-7B) | 18.6K | 38.7% |
> | GUI-Cursor-7B | 8K | 56.5% |
> ||
>
> Some models use UI-TARS-1.5-7B as the base model, which is fine-tuned from Qwen2.5-VL-7B with massive training data/ We compare the previous leading mode GTA1 and the concurrent work GUI-Spotlight:
>
> | Model | # Training Samples | ScreenSpot-Pro Acc. |
> | :--- | :--- | :--- |
> | UI-TARS-1.5-7B | - | 38.7%|
> | GTA1-7B | 64K | 50.1% |
> | GUI-Spotlight-7B (UI-TARS-1.5-7B) | 18.6K | 52.8% |
> ||
>
> Regarding the efficiency of multi-turn RL, the average training time per step for one-step and four-step movements is 289 and 680 seconds, respectively. The primary bottleneck is online trajectory rollout, which increases from 157 to 432 seconds when using four-step movement. _However, our method remains more efficient overall, achieving better accuracy with significantly fewer samples and fewer total steps._
>
>
> **Question 3**
> > Experiments are conducted only on GUI grounding benchmarks. It remains unclear whether the proposed method generalizes to broader GUI agent tasks—especially those involving user instructions that do not exactly match GUI elements. Additional experiments on datasets such as AITW or Multimodal-Mind2Web would strengthen the claim that the proposed interactive reasoning approach can handle more realistic and diverse scenarios.
>
> Thank you for raising this – after your question, we started the online evaluation on OSWorld and the offline evaluation on Multimodal-Mind2Web. For OSWorld, we use the same planning method as GTA1 [1], allowing us to directly compare our method with it. For Multimodal-Mind2Web, we use the existing planning results from UGround [2] to fairly compare grounding performance. We are currently running these experiments; we will notify you as soon as they finish and include the results in the updated version of the paper.
>
> [1] GTA1: GUI Test-time Scaling Agent (https://arxiv.org/abs/2507.05791)
>
> [2] Navigating the Digital World as Humans Do: Universal Visual Grounding for GUI Agents (https://arxiv.org/abs/2410.05243, ICLR’25)

---

> ### Author Response · Authors · 2025-11-21
>
> **Question 4**
>
> > It is unclear whether reinforcement learning is necessary for this method. Could the proposed approach also function effectively without additional training, for example in a test-time scaling or supervised fine-tuning (SFT) setting?
>
> _Why use RL rather than SFT_
>
> In GUI-Cursor, the model controls a cursor to locate targets through multiple cursor movements, and reinforcement learning is a natural framework for modelling this sequential decision-making process.
>
> We tried SFT in our preliminary experiments and found it costly and less effective. SFT requires large-scale ground-truth cursor-movement trajectories, which we synthesised using GPT-4o and filtered out unsuccessful locating samples. However, generating training trajectories was expensive due to the multi-turn nature of the task and the low success rate of large models in cursor control. Qwen2.5-VL-7B trained on 10K synthesised trajectories achieved only 49.6% accuracy on ScreenSpot-v2, whereas our RL method achieved 93.9% with significantly less engineering effort.
>
> While SFT is often necessary to address the cold-start problem (e.g., learning output formats), our base model achieves sufficient initial accuracy in multi-step cursor control (36.3% on ScreenSpot-v2, shown in Figure 5), allowing us to apply RL directly without additional effort.
>
> _Test-time scaling without fine-tuning_
>
> The experimental results of test-time scaling without fine-tuning are shown in Figure 5 and Table 3. This strategy requires the model to use its inherent spatial reasoning abilities to identify the spatial relations between the cursor and the target. We found that GPT-4o can benefit from this strategy (17.5% -> 25.5% with 10 steps relative movement) while Qwen2.5-VL-7B does not (88.8% -> 36.3% with 4 steps direct movement).

---

> ### Author Response · Authors · 2025-11-26
>
> **Question 3 Update**
>
> We sincerely thank the reviewer for the suggestion to evaluate on agentic benchmarks. To demonstrate that GUI-Cursor generalises to realistic and diverse scenarios beyond grounding benchmarks, we have conducted two additional evaluations: 1) Online evaluation on OSWorld and 2) Offline evaluation on Multimodal-Mind2Web. The detailed results are as follows:
>
> ### OSWorld
>
> We use the same o3-based planning strategy used by GTA1-7B for fair comparison. The results shown in the table below indicate that GUI-Cursor significantly improves agent performance. Using fewer action steps, GUI-Cursor-7B (57.1% at 50 steps) outperforms GTA1-7B (53.1% at 100 steps), and also outperforms the larger model GTA1-32B (55.4% at 100 steps). These results demonstrate that GUI-Cursor can handle more realistic grounding tasks in agentic scenarios.
>
> | Agent Model | Acc. (15 steps)| Acc. (50 steps)| Acc. (100 steps) |
> |-|-|-|-|
> | o3 | 9.1% | 17.2% | 23.0% |
> | UI-TARS-1.5-7B | - | - | 27.4% |
> | GTA1-7B w/ o3 | - | - | 53.1% |
> | GTA1-32B w/ o3 | - | - | 55.4% |
> | GUI-Cursor-7B w/ o3 | - | **57.1%** | - |
> ||
>
> ### Multimodal-Mind2Web
>
> We use the planning results provided by UGround to compare the grounding performance fairly. As shown in the table (below), GUI-Cursor achieves the highest average accuracy, but we observe that performance differences are small across all models for this benchmark. Furthermore, we manually checked the results and found that the majority of errors stem from the planner (GPT-4o) rather than the grounding module. So to improve results, one should use a better planner such as GPT-5.
>
> | Method | Cross-Domain Acc. | Cross-Task Acc. | Cross-Website Acc. | Avg Acc. |
> |-|-|-|-|-|
> | UGround-V1-7B | 48.5% | **50.7%** | 48.1% | 49.1% |
> | GUI-G2 | 48.5% | 49.1% | 48.2% | 48.6% |
> | GTA1-7B | 48.7% | 49.3% | 47.9% | 48.6% |
> | GUI-Cursor-7B | **48.9%** | 50.2% | **49.0%** | **49.4%** |
> ||
>
> In summary, the results on the agentic benchmarks OSWorld and Multimodal-Mind2Web demonstrate that GUI-Cursor can handle grounding tasks in these more realistic and diverse scenarios.

---

### Comment · Area_Chair_NzSZ · 2025-11-25

Dear Reviewers,

A quick reminder that the authors have posted their responses. The discussion period ends on December 2, so please review the rebuttal and share any follow-up comments as soon as possible. Your timely input is greatly appreciated. Thanks.

Best,

Your ACs

---

### Author Response · Authors · 2025-11-27
**Global Response [1/2]**

Thank you for the thoughtful reviews and constructive follow-up questions. We’re excited that reviewers highlighted our work is **well-motivated**,  **solid**, **simple and effective**, and  **achieves strong empirical gains**. Below we summarise cross-review updates, new experiments, and clarifications added during the discussion process.

---
## 1. Additional Experiments
- **Generalisation of Spatial Reasoning**, requested by 1NKj (spatial reasoning scope), opuB (cursor-in-box is simplified), JBDJ (cursor‑in‑box probe is neat but synthetic).
    We ran additional experiments on two spatial reasoning benchmarks SpatialMQA and SPHERE (suggested by reviewer JBDJ). The results show that GUI-Cursor improves performance by +5.3 on SpatialMQA and +1.8 on SPHERE, with the largest gains in “Reasoning” (+1.9) and “Combine 2 Skills” (+3.9) categories. This confirms that _GUI-Cursor obtain better spatial reasoning that can generalise to out-of-distribution domains_.
- **Online Evaluation**, requested by 1NKj, MH8z, and opuB (evaluation in agent settings).
    We ran additional experiments on OSWorld. Using the same planner, GUI-Cursor-7B reaches 57.1% at 50 steps, yielding more accurate results than the prior SOTA GTA1-7B (53.1%) and GTA1-32B (55.4%) at 100 steps. This confirms _GUI-Cursor can handle more realistic and diverse scenarios_.
- **Grounding Evaluation**, requested by opuB (beyond the ScreenSpot family).
    Additional evaluations on UI-Vision and OSWorld-G show that GUI-Cursor consistently yields significant improvements. With Qwen2.5-VL-7B as the base model, results on UI-Vision improve from 0.9% to 25.7%, and OSWorld-G from 31.4% to 58.0%; with UI-TARS-1.5-7B as the base model, results on UI-Vision improve to 27.3% and OSWorld-G to 65.6%. _The substantial improvements in “Spatial” and “Manipulation” categories further demonstrate the benefit of our task reframing_.
- **Additional Ablations**, requested by 1NKj (single-step vs multi-step training).
    After training with `max_steps=1`, the accuracy of GUI-Cursor ScreenSpot-Pro drops from 56.5% to 47.8% (and 45.3% $\to$ 40.7% without CCF), _providing additional evidence on the effectiveness of multi-step and visual feedback during training_.

## 2. Positioning and Novelty

**Comparison with Concurrent Work**, requested by JBDJ (iterative crop/zoom RL), 1NKj (newer leaderboard entries), and MH8z (novelty beyond visual feedback).
    Suggested by reviewers, we include a discussion on concurrent works: GUI-Spotlight, GUI-ARP, and GUI-RC/RCPO. _These approaches primarily focus on improving grounding accuracy by narrowing the region via iterative cropping_. GUI-Cursor is a complementary method for the following reasons:

1. **Task Reframing:** GUI-Cursor reframes GUI grounding as a continuous cursor-controlling task, and proposes a set of trajectory penalties tailored to cursor control.
2. **Objective:** GUI-Cursor’s primary objective is to alleviate the spatial semantic alignment problem, which is achieved by reasoning over visual feedback from cursor usage.
3. **CCF is Simple and Effective**: CCF is introduced to balance training efficiency and inference accuracy, and cropping is only applied at inference time. In our experiments, we found that this cropping strategy can also be easily applied to other baselines (GTA1 and GUI-G2), significantly improving their accuracy.

The concurrent works primarily focus on better iterative cropping strategies for reducing the difficulty of predicting coordinates in GUI images. We included a discussion about these works compared to CCF in the revised paper:

- GUI-Spotlight (released on Oct. 5th, 2025) introduces three tools for narrowing the GUI region iteratively, and is trained via RL. CCF is simpler and more effective; for example, applying CCF to GTA1 yields better results than GUI-Spotlight on ScreenSpot-Pro by 2.2% without any further training.
- GUI-ARP (released on Sep. 19th, 2025) proposes a cropping strategy by considering the signals from attention scores. Though it shows improvements on the higher resolution GUIs, it is less effective in tasks with standard image sizes (e.g., it only obtains 91.8% in ScreenSpot-v2). A possible explanation is that it does not address the underlying spatial-semantic alignment issue.
- GUI-RC/RCPO (released on Aug. 8th, 2025) samples multiple predictions and uses voting or test-time RL to improve the grounding accuracy. GUI-RC also does not require training, but CCF remains simpler and more effective; for example, on ScreenSpot-Pro, GUI-G2-7B +CCF achieves higher accuracy (52.1%)  compared to GUI-G2-7B +GUI-RC (47.9%).

GUI-Cursor is a complementary approach, and better cropping methods could be integrated to further improve performance. However, CCF remains the simplest and most effective among these concurrent methods. *We also note that these concurrent works are not required to be included according to the ICLR 2026 Reviewer Guide.*

---

### Author Response · Authors · 2025-11-27
**Global Response [2/2]**

## 3. Efficiency and Practicality

- **Training and Data Efficiency**, requested by MH8z.
    We explain GUI-Cursor’s data efficiency in our answer. GUI-Cursor only uses 8K samples compared to GUI-Actor-7B (1.76M samples), GUI-G2-7B (100K), and GTA1-7B (64K). In our analysis, we show that the multi-step rollout is the main efficiency bottleneck; however, GUI-Cursor remains the most efficient overall, achieving better accuracy with significantly fewer samples and fewer total steps. Appendix F in the updated version of the paper analyses the training and data efficiency of GUI-Cursor.
- **Inference Efficiency**, requested by opuB (context growth and latency) and 1NKj (overhead introduced by the extra steps).
    In our experiments, the throughput is 2.35/2.08/1.38 samples per second for 1/2/3 steps; since 95% of rollouts terminate within two steps, the overall overhead introduced by GUI-Cursor is ~12.5% compared to vanilla single-step inference. (opuB agrees it is acceptable given the performance gains)
- **Why RL over SFT/Test-Time Scaling**, requested by MH8z.
    SFT on 10K GPT-4o synthesised trajectories with filtering reached only 49.6% on ScreenSpot-v2 and was costly to curate; RL directly on the base model reached 93.9% without extensive engineering efforts. Test-time scaling requires VLMs to possess strong initial spatial reasoning capabilities: it helps GPT-4o modestly but harms Qwen2.5-VL-7B, which is discussed in the original submission.


## 4. Submission Revision

We have extended the submission to 10 pages with the following modifications:

- Figure Correction: We fixed Figure 2(b) to eliminate overlapping labels (suggested by 1NKj).
- New Benchmarks: We incorporated results from two additional grounding benchmarks, the OSWorld online evaluation, two spatial reasoning benchmarks, and a new base model.
- Efficiency Analysis: We added the analysis regarding training efficiency and inference throughput.
- Discussion on Concurrent Works: We included the discussion on concurrent works (GUI-Spotlight, GUI-ARP, GUI-RC/RCPO).

---

We appreciate the reviewers’ insights. The added experiments demonstrate that GUI-Cursor improves grounding, agent performance, and spatial reasoning while remaining efficient in terms of data and compute.

---

### Meta-Review · Area_Chair_yxtv · 2026-01-07

**Summary:**

This paper recieves 2x BA and 2x BR, where core concerns are novelty and effectiveness compared with recent cropping-based methods (e.g., GUI-ARP), generalization on spatial reasoning, and evaluation in agentic settings.

After checking all reviews and responses, the AC recommend rejection. While the cursor-driven interaction is a plausible direction, the current work does not provide compelling evidence of its superiority over existing crop-based approaches, espacially GUI-ARP.

Experimental results confirm that GUI-Cursor lags behind the crop-based GUI-ARP in core performance metrics. GUI-ARP (GUI-Actor-7B) achieves 60.8% accuracy on ScreenSpot-Pro, while the strongest variant of GUI-Cursor (UI-TARS-1.5-7B) only reaches 58.1%. This performance gap directly undermines the paper’s claim that cursor-driven interaction is a superior alternative to cropping-based strategies.

**Reviewer Concerns:**

Review Concerns:

Reviewer MH8z

W1. [Not addressed] About the novelty in the broader vision domain.
The authors request specific related literatures but the reviewer did not reply.

W2. [Partially addressed] About the training efficiency and performance.
The authors only provide training samples, not efficiency. Actual training costs are not specifically compared.

W3. [Addressed] Generalization to broader GUI agent tasks.

Q1. [Addressed] About the RL stage.

Reviewer 1NKj

W1. [Addressed] Broader GUI agent systems evaluation.

W2. [Addressed] About the title and the scope.

W3. [Addressed] About the computational cost.

Q1. [Addressed] Single-step ablation.

Q2. [Addressed] Evaluation in Agent settings.

Q3. [Not Fully Addressed] Comparison with related works.
The authors provide a brief discussion on crop-based related works, but fail to demonstrate the unique advantage of the proposed cursor-based method with other crop-based methods.

Q4. [Addressed] Figure presentation.

Reviewer opuB

W1. [Addressed] Broader evaluation

W2. [Addressed] Inference Efficiency

W3. [Addressed] About broader spatial reasoning beyond GUI.

Reviewer JBDJ

W1. [Not addressed] Comparison with related works.
The authors provide a brief discussion on crop-based related works, but fail to demonstrate the unique advantage of the proposed cursor-based method with other crop-based methods.

W2. [Partially addressed] Outdated SOTA performance.
The authors acknowledge it but said it was cocurrent work, and did not provide any further evidence to verify the unique advantage of the proposed cursor-based method.

W3. [Addressed] Missing related works.

W4. [Addreseed] Evaluation on general spatial reasoning benchmarks.

**Reviewer Scores:**

Reviewer Scores

For two negative reviewers:

- Reviewer MH8z scores 4 and may not upgrade his rating as the authors fail to sufficiently explain its novelty.

- Reviewer JBDJ scores 4 and may upgrade his rating as the authors have demonstrate the effectiveness on general spatial reasoning. However, there is possible for the reviewer to maintain the negative score as this paper is actually outdated sota performance.

---

### Decision · Program_Chairs · 2026-01-26

Reject